# Chronic spindle assembly checkpoint activation causes myelosuppression and gastrointestinal atrophy

Gerlinde Karbon [ID][1], Fabian Schuler [ID][1], Vincent Z Braun [ID][1], Felix Eichin [ID][1], Manuel Haschka[1], Mathias Drach[2], Rocio Sotillo [ID][3], Stephan Geley[4], Diana CJ Spierings [ID][5], Andrea E Tijhuis [ID][5], Floris Foijer [ID][5] & Andreas Villunger [ID][1,6 ✉]

## Abstract

**Interference with microtubule dynamics in mitosis activates the spindle assembly checkpoint (SAC) to prevent chromosome segregation errors. The SAC induces mitotic arrest by inhibiting the anaphase-promoting complex (APC) via the mitotic checkpoint complex (MCC). The MCC component MAD2 neutralizes the critical APC cofactor, CDC20, preventing exit from mitosis. Extended mitotic arrest can promote mitochondrial apoptosis and caspase activation. However, the impact of mitotic cell death on tissue homeostasis in vivo is ill-defined. By conditional MAD2 overexpression, we observe that chronic SAC activation triggers bone marrow aplasia and intestinal atrophy in mice. While myelosuppression can be compensated for, gastrointestinal atrophy is detrimental. Remarkably, deletion of pro-apoptotic *Bim/Bcl2l11* prevents gastrointestinal syndrome, while neither loss of *Noxa/ Pmaip* or co-deletion of *Bid* and *Puma/Bbc3* has such a protective effect, identifying BIM as rate-limiting apoptosis effector in mitotic cell death of the gastrointestinal epithelium. In contrast, only overexpression of anti-apoptotic BCL2, but none of the BH3-only protein deficiencies mentioned above, can mitigate myelosuppression. Our findings highlight tissue and cell-type-specific survival dependencies in response to SAC perturbation in vivo.**

**Keywords** Spindle Assembly Checkpoint; MAD2; Mitosis; BH3-only Proteins; Apoptosis
**Subject Categories** Autophagy & Cell Death; Cell Cycle; Signal Transduction

## Introduction

During mitosis, genetic information needs to be propagated evenly between daughter cells. Mistakes in chromosome segregation and subsequent aneuploidy can harm cell fitness and organismal health (Holland and Cleveland, 2009; Tang and Amon, 2013). Several checkpoints monitor this process to regulate the fidelity of cell division. During mitosis, the spindle assembly checkpoint (SAC) is critical for adequately segregating sister chromatids into daughter cells (Bharadwaj and Yu, 2004; Musacchio and Salmon, 2007). The main executioner of the SAC, the mitotic checkpoint complex (MCC), consists of Budding Uninhibited By Benzimidazoles 1 Homolog Beta (BUB1B/BUBR1), Budding Uninhibited By Benzimidazoles 3 Homolog (BUB3), Cell Division Cycle 20 Homolog (CDC20) and Mitotic Arrest Deficient 2 (MAD2) (Alfieri et al, 2016; Musacchio, 2015). During spindle formation in early mitosis, MAD2, encoded by *MAD2L1*, is conformationally activated by kinetochore-bound Mitotic Arrest Deficient 1 (MAD1), which converts the inactive soluble open MAD2 protein conformation into its active closed form, able to bind and neutralize CDC20. The MCC then binds to and prevents the full activation of the Anaphase-Promoting Complex (APC), which is required for the ubiquitination-initiated degradation of securin and cyclin B1, which are required for sister chromatid separation and exit from mitosis, respectively. The SAC becomes inactivated upon bipolar chromosome attachment, which causes the depletion of MAD1–MAD2 from properly microtubule-attached kinetochores. Thus, the SAC monitors spindle formation and keeps cells in prometaphase until all chromosomes are correctly attached to the mitotic spindle (Alfieri et al, 2016; De Antoni et al, 2005; Kulukian et al, 2009; Mapelli et al, 2007).

Cell culture experiments suggest that mitotic cell death can be triggered if mitosis is prolonged by preventing SAC satisfaction in due time (Rieder and Maiato, 2004). Alternatively, cells may escape prolonged mitosis by a process called "mitotic slippage" or "checkpoint adaptation", triggered by the gradual noncanonical

[1]Institute for Developmental Immunology, Biocenter, Medical University of Innsbruck, Innsbruck, Austria. [2]Dermatology, General Hospital, University Hospital Vienna, Vienna, Austria. [3]German Cancer Research Center (DKFZ), Division of Molecular Thoracic Oncology, Heidelberg, Germany. [4]Institute for Pathophysiology, Biocenter, Medical University of Innsbruck, Innsbruck, Austria. [5]European Research Institute for the Biology of Ageing, University of Groningen, University Medical Center Groningen, 9713 AV Groningen, The Netherlands. [6]CeMM Research Center for Molecular Medicine of the Austrian Academy of Sciences, 1090 Vienna, Austria. ✉E-mail: andreas.villunger@i-med.ac.at

degradation of cyclin B by the APC/C, allowing cell survival (Brito and Rieder, 2006; Gascoigne and Taylor, 2008; Haschka et al, 2018; Topham and Taylor, 2013). Slippage is frequently associated with increased ploidy and deregulated centrosome numbers, causing a substantial fraction of these cells to undergo senescence or cell death (Godinho et al, 2009; Sinha et al, 2019; Yamada and Gorbsky, 2006). However, some poly- or aneuploidy cells can thrive and re-enter the cell cycle. This is facilitated by centrosome clustering, particularly when the function of the *p53* tumor suppressor gene is impaired or lost (Aylon et al, 2006; Senovilla et al, 2009). Such features enable chromosomal instability (CIN) and aneuploidy and are frequently found in cancer (Santaguida and Amon, 2015).

Complete loss of MAD2, or other MCC proteins leading to SAC deficiency, are embryonic lethal in mice, while hypomorphic alleles can cause microcephaly, premature ageing phenotypes and cancer predisposition syndromes (Dobles et al, 2000; Iwanaga et al, 2007; Wang et al, 2004). To name a few examples, mutations of the SAC components cause mosaic-variegated aneuploidies (Hook and Warburton, 2014), with germline mutations in *BUB1* linked to microcephaly (Carvalhal et al, 2022) and mutations in *MAD1L1* causing multiple malignancies (Villarroya-Beltri et al, 2022). Together, these findings underpin the importance of the mitotic spindle assembly checkpoint in development and tumor suppression. Of note, even partial loss of MAD2 function leads to premature degradation of securin and separation of the sister chromatids, causing aneuploidy and polyploidy in cell lines and animal models. Consistently, haploinsufficiency accelerates T-cell lymphomagenesis in a *p53*-deficient background and lung cancer progression in preclinical cancer models (Michel et al, 2001).

Consistent with the mode of MAD2 action, overexpression can hyperactivate the SAC and lead to polyploidy, lagging chromosomes, and aneuploidy (Hernando et al, 2004; Sotillo et al, 2007; Sotillo et al, 2010). Consequently, tumourigenesis was accelerated by MAD2 overexpression together with MYC in blood cells or mutated KRAS in the lung epithelium (Sotillo et al, 2007; Sotillo et al, 2010). However, upon MAD2 overexpression, increased cell death rates were also noted, a phenomenon recently linked to Forkhead Box M1 (FOXM1) transcription factor expression levels (Pan et al, 2023). This may account for delayed tumor onset reported upon MAD2 overexpression in a *Her2*-driven mouse model of breast cancer (Rowald et al, 2016). While SAC genes, including *MAD2L1*, are rarely found mutated in human cancers, overexpression is observed more frequently (summarized by Simonetti et al (2019)). Besides breast cancer, increased MAD2 protein expression correlates with increased mortality and cancer recurrence in humans (Byrne et al, 2017).

A series of in vitro studies using model cell lines have shown that the BCL2 protein family is critically involved in regulating mitotic cell death and cell death after slippage (Haschka et al, 2018; Topham and Taylor, 2013). The B-Cell CLL/Lymphoma 2 (BCL2) family comprises a set of anti-apoptotic molecules, including BCL2 itself, BCL2 like 1 (BCL2L1/BCLX) or Myeloid Cell Leukemia Sequence 1 (MCL1), as well as BH3-only proteins that act as pro-apoptotic stress sensors, e.g., BCL2-Interacting Mediator of Cell Death (BIM/BCL2L11), BH3 Interacting Domain Death Agonist (BID) and Phorbol-12-Myristate-13-Acetate-Induced Protein 1 (PMAIP/NOXA), along with apoptotic effectors, BCL2 Associated X Protein (BAX) and BCL2 Antagonist/Killer 1 (BAK1), that are required for pore-formation at the outer mitochondrial membrane,

kick-starting a pro-apoptotic signaling cascade (Bleicken et al, 2014; Salvador-Gallego et al, 2016). During severe mitotic delays, the pro-survival BCL2-family member MCL1 acts as a molecular timer, where its NOXA-dependent degradation facilitates BIM-induced cell death in cancer cell lines (Haschka et al, 2015). In addition, the BH3-only proteins BID (Wang et al, 2014) and Bcl2 Modifying Factor (BMF) (Pan et al, 2023) have been implicated in mitotic cell death and may act complementary to BIM and/or NOXA in contexts or cell types that still need to be defined. Loss of MCL1 expression renders epithelial cancer cells highly dependent on anti-apoptotic BCLX. This represents a therapeutic vulnerability that can be targeted by so-called "BH3 mimetics" that exploit the mode of action of BH3-only proteins, leading to BAX/BAK activation (Bennett et al, 2016; Shah et al, 2010). Notably, mRNA expression of the BH3-only protein *NOXA* can be an independent survival predictor in human breast cancer patients treated with microtubule-targeting agents (Karbon et al, 2021).

While compelling, whether the same molecular players mediating cell death in response to spindle poisons in cancer cell lines ex vivo are also critically involved in the cellular response to chronic SAC deregulation in vivo was, to the best of our knowledge, never tested. Hence, we exploited a mouse model allowing conditional over-expression of the MCC component MAD2 across tissues (Rowald et al, 2016). Using this system, we aimed to address whether mitochondrial apoptosis limits the survival of cells experiencing chronic SAC activity and potentially CIN-related pathology.

# Results

## MAD2 overexpression triggers mitotic delay and cell death in hematopoietic cells

First, we assessed the impact of aberrant MAD2 expression on the survival of highly proliferative cells, focusing on the blood and expecting hematopoietic progenitor cells to be highly vulnerable to SAC perturbation. Herein, we used mouse bone marrow, immortalized with the homeobox B8 gene, *HoxB8* (Redecke et al, 2013; Wang et al, 2006), to create SCF-dependent myeloid progenitors of neutrophils (dubbed PN) or multi-potent progenitor cells that depend on FLT3 ligand (dubbed PF), respectively. HoxB8-PN or HoxB8-PF cells were derived from mice allowing over-expression of MAD2 upon doxycycline (Dox) addition. In this model, N-terminally HA-tagged murine *Mad2L1* was knocked into the *Col1a* locus and controlled by the reverse tetracycline transactivator (rtTA) expressed from the *Rosa26* locus. This allows near-ubiquitous transgene expression in response to Dox treatment (Rowald et al, 2016). In addition, we crossed-in a human BCL2 transgene, driven by the *Vav*-gene promotor, active in all hematopoietic cells (Ogilvy et al, 1999). Transgenic animals and bone marrow-derived cell lines are referred to as: R (*R26rtTA*), MR (*Mad2/R26rtTA*), or MR2 (*Mad2/R26rtTA/BCL2*) throughout the text and figures. In some settings, these mice or cell lines also carried a Dox-responsive GFP reporter (G), targeted to the second *Col1a* allele (Premsrirut et al, 2011). The *HA-Mad2* allele was maintained in a hemizygous state. Hence, as an example, primary MRG2-derived bone marrow cells harbor four transgenes (*Mad2/R26rtTA/GFP/BCL2*) that were eventually immortalized using a HoxB8-encoding retrovirus (Schuler et al, 2019).

Then, we induced MAD2 expression by Dox treatment in vitro and monitored transgene levels over time using an anti-HA antibody. Bone marrow-derived cell lines, MR and MR2, showed increasing MAD2 levels in a dose- and time-dependent manner (Figs. 1A and EV1A). Constitutive overexpression of transgenic *BCL2* was confirmed in MR2 cells using a human BCL2-specific antibody, while endogenous BCL2, BCLX and MCL1 levels were comparable between both genotypes (Figs. 1A and EV1A,B). As noted before, high human BCL2 levels also allowed cells to tolerate higher levels of BIM (Fig. 1A; Bouillet et al, 1999; O'Connor et al, 1998). Interestingly, BIM levels dropped in response to Dox treatment-induced MAD2 expression, best visible in BCL2 transgenic cells after 8 and 12 h (Fig. 1A). This likely reflects its increased turnover by the APC in cells experiencing mitotic delay or arrest (Fava et al, 2013; Wan et al, 2014).

Next, we conducted immunoprecipitation experiments to demonstrate that the increase in MAD2 ends in increased levels of active closed MAD2 (De Antoni et al, 2005; Mapelli et al, 2007). Using a conformation-specific MAD2 antibody, we confirmed excess active closed MAD2 upon Dox addition (Fig. 1B). Increased levels of closed MAD2 showed a strong correlation with a higher percentage of phospho-histone H3 positive (phH3 + ) HoxB8-PN cells. This correlation was monitored using flow cytometry analysis and is indicative of mitotic delay or arrest in these cells (Figs. 1C and EV1C). Viability analysis showed an increase in the percentage of MR cells being propidium iodide positive (PI$^+$), indicating increased levels of cell death in cultures upon Dox-induced MAD2 overexpression (Figs. 1D and EV1D). The percentage of PI$^+$ cells was strongly reduced in the presence of the BCL2 transgene (Figs. 1D and EV1D). Time-lapse microscopy analysis of MR and MR2 cells cultured in the presence of PI underlined the beneficial effect of BCL2 overexpression (Fig. EV1E). However, this effect vanished over time, indicating secondary necrosis in culture (Fig. 1D).

We also exploited the Dox-responsive Green Fluorescent Protein (GFP)-reporter as a surrogate marker of MAD2 expression in living cells by flow cytometry. GFP expression was readily detectable by flow cytometry analysis and on the protein level by immunoblotting (Fig. EV1F,G). We exploited this system to perform a competition assay comparing MAD2-transgenic cells that turn green upon Dox addition (MRG) and rtTA single transgene-positive control cells (R), expressing neither the GFP reporter nor MAD2. In MRG cells, increased GFP expression correlated with increased cell death, as measured by PI uptake and BCL2 overexpression again blocked cell death in MRG2 cells (Fig. 1E). When MRG cells were mixed in a 1:1 ratio with control cells (R), we observed that the latter rapidly outcompeted the MAD2 overexpressing cells, as inferred by the rapid plateau in viable GFP$^+$ cells (Fig. 1F). Notably, BCL2 transgenic cells failed to perform substantially better in this competition assay, indicating they may no longer proliferate at comparable rates (Fig. 1F). Importantly, GFP expression alone was not toxic, as assessed by flow cytometric analysis of RG cells placed on Dox (Fig. EV1H). In order to directly correlate GFP levels to MAD2 expression, we also isolated three populations of MRG cells after Dox addition. GFP-negative, GFP-low and GFP-high cells were subjected to western blot analysis. This revealed that while the levels of GFP monitored by flow cytometry correlated well with protein levels detected by western, this was not the case for MAD2. In fact, while GFP-low

and GFP-high cells had comparable MAD2 expression levels, GFP-negative cells showed some transgene expression upon longer exposure. In contrast, this was not seen for GFP (Fig. EV1). Together, this suggests that we actually may underestimate the percentage of MAD2 overexpressing cells in our flow cytometric analyses, as apoptosis induction also causes loss of GFP expression.

Next, we performed live-cell imaging to better understand cell fate in the presence of BCL2. We could observe that MAD2 overexpressing cells spend more time in mitosis than non-induced cells, ending primarily in mitotic cell death without cytokinesis (Fig. 1G,H). As expected, cells arrested largely at metaphase, indicating SAC hyperactivation despite successful spindle formation. The duration of the mitotic arrest was not altered once BCL2 was overexpressed. Importantly, however, we noticed that the presence of BCL2 reduced death in mitosis (diM) from about 60% to 20% in MR2 cells, paralleled by a significant increase in cells managing mitotic (M)-exit (Figs. 1I; Movies EV1–4).

As we noted reduced cell death and increased mitotic exit rates in BCL2 transgenic cells, we hypothesized that such cells may display CIN. Therefore, we performed single-cell whole-genome sequencing (scWGS) at different time points after induction of MAD2 overexpression. Flow cytometry was used to confirm the effects of MAD2 overexpression on cell cycle progression via phH3 staining and cell survival using sub-G1 analysis before scWGS (Fig. EV2A,B). Surprisingly, we neither detected karyotypic abnormalities in the absence or presence of Dox in R, R2, MR, or MR2 cells analyzed after 24 h, nor within MR2 cells still alive after 3 or 7 days (Fig. EV2C,D). This indicates that MR cells die in mitosis; those that survive due to exogenous BCL2 overexpression eventually must manage to properly terminate SAC signaling, as more cells exited mitosis (Fig. 1I) and we failed to observe aneuploidy (Fig. EV2C,D), despite MAD2 overexpression.

In summary, overexpression of MAD2 leads to mitotic delay and cell death in *HoxB8*-immortalized hematopoietic progenitor cells. However, MAD2 overexpression does not cause viable karyotypic changes in tissue culture. The pro-survival protein BCL2 can reduce cell death induced by MAD2 overexpression. This may give cells enough time to satisfy the SAC and complete mitosis, supported by the fact that no karyotypic abnormalities were noted in scWGS analyses.

## Systemic overexpression of MAD2 can cause premature lethality

Next, we aimed to interrogate the consequences of MAD2 overexpression on the hematopoietic system in vivo by putting R and MR mice on Dox-containing diet. Unexpectedly, we noted that transgenic mice rapidly lost body weight and 4/6 animals became clearly moribund within 5 days of Dox treatment and had to be sacrificed (Fig. 2A). Bone marrow cellularity, however, was not changed within this period (Fig. 2B). A time-dependent increase of phH3$^+$ cells within the bone marrow was noted from day 1 onwards, documenting SAC activation and indicating mitotic delays upon MAD2 overexpression (Fig. 2C). Along this line, we detected increased cell death in bone marrow from day 3 onwards (Fig. 2D). Curiously, the percentage of Lin$^-$Sca1$^+$cKit$^+$ (LSK) cells increased significantly upon MAD2 overexpression (Fig. 2E,F). We also noted an accumulation of common lymphoid and common myeloid progenitors (CLP/CMP) upon SAC perturbation on day 5

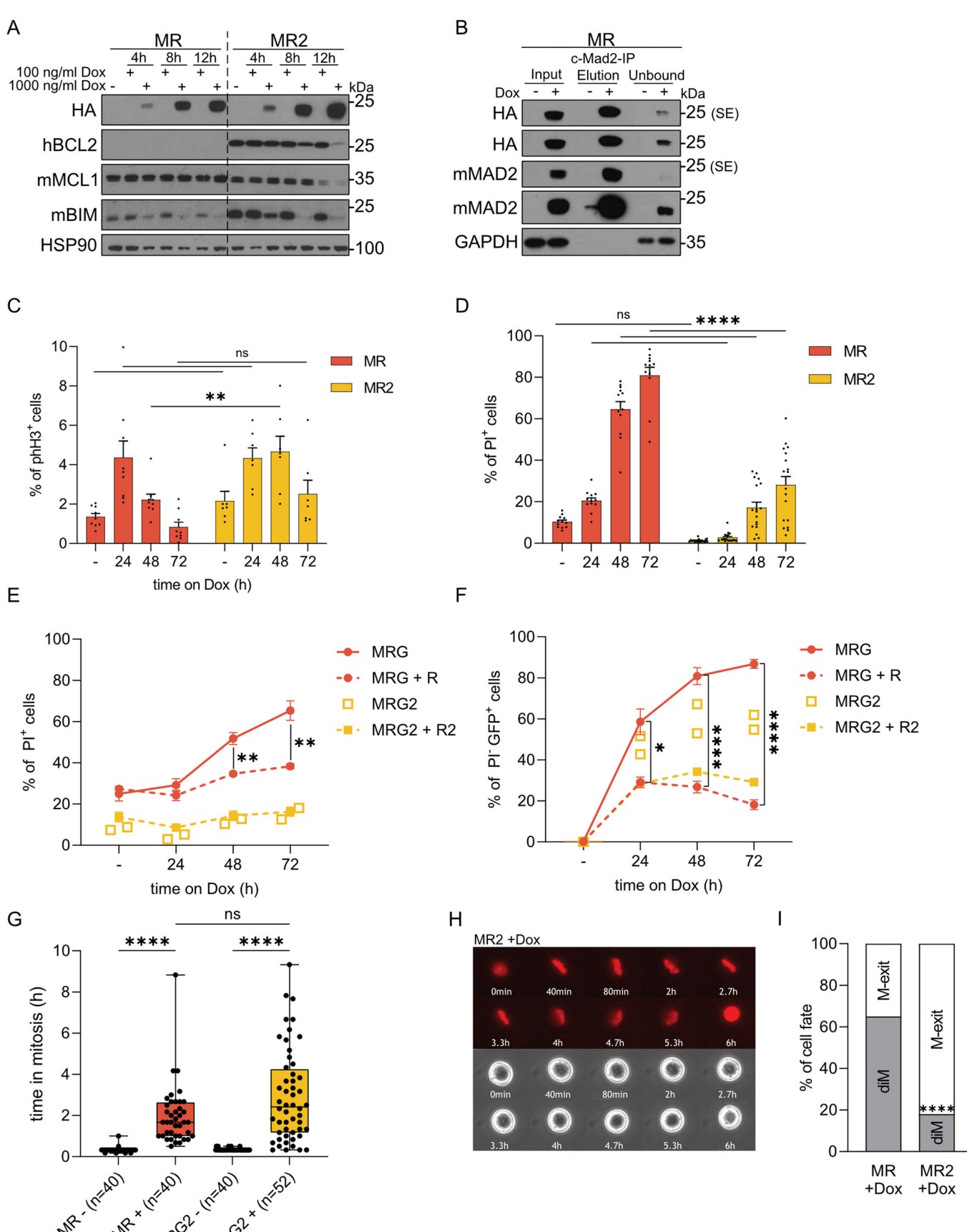

◀   **Figure 1.   MAD2 overexpression triggers mitotic delays and BCL2-regulated cell death.**

HoxB8 cells were generated from the bone marrow of mice of the indicated genotypes and treated with Doxycycline for the indicated time points or 24 h, if not stated otherwise. (**A**) HoxB8 PNs were harvested for immunoblot analysis with indicated antibodies after Mad2 transgene induction with 100/1000 ng/ml Doxycycline. (**B**) Input, elution and unbound fractions from immunoprecipitation analysis to detect MAD2 in its closed (active) conformation. (**C**) HoxB8 PNs were fixed with ethanol and analyzed by flow cytometry to quantify cells in mitosis staining positive for phH3$^+$; MR ($n = 9$; 6 biological and 1–2 technical replicates), MR2 ($n = 7$; 3 biological and 1–3 technical replicates); mean ± SEM. (**D**) HoxB8-PF cells were stained with PI and viability was analyzed by flow cytometry: MR ($n = 12$; 6 biological and 1–3 technical replicates), MR2 ($n = 18$; 7 biological and 2–3 technical replicates); mean ± SEM. (**E, F**) HoxB8 PNs were cultured alone or mixed 50:50 with transgene-negative cells in the presence or absence of Doxycycline and analyzed for viability (PI$^+$) and GFP expression by flow cytometry; MRG ($n = 6$; 5 biological and 1–2 technical replicates), MRG + R ($n = 11$; 5 MRG cell lines mixed with 6 R cell lines), MRG2 ($n = 2$, 2 biological replicates) MRG2 + R2 ($n = 6$; 2 MRG2 cell lines mixed with 3 R2 cell lines); mean ± SEM. (**G**) HoxB8 PF cells were subjected to live-cell imaging to evaluate time spent in mitosis, from chromosome condensation until change of cell fate, with or without MAD2 induction. Min to max with median and IQR of: MR- 0.55 MR + 0.81, MRG- 0.55, MRG2 + 0.51; single experiment, randomly selected cells in different positions; MR-Dox ($n = 40$), MR+Dox ($n = 40$), MR2-Dox ($n = 40$), MR2+Dox ($n = 52$); (**H**) Representative live-cell imaging time series of an MR2 transgenic cell after MAD2 induction, eventually dying in mitosis (DNA stained in red with SPY650-DNA & phase contrast). (**I**) Cell fate of cells shown in (**G**), diM = death in mitosis, M-exit = cells exiting mitosis. Data information: (**C–F**) Two-way ANOVA, Šídák's multiple comparisons. (**G**) Kruskal–Wallis test, Dunn's multiple comparison. (**I**) unpaired *t* test. ns not significant, *$P < 0,05$, **$P ≤ 0.01$, ****$P ≤ 0.0001$.

(Appendix Fig. S1A,B). The accumulation of LSK, CLP and CMP cell fractions in the bone marrow suggested delayed maturation or increased mobilization due to the loss of more mature blood cell types. This aligned well with the significantly reduced rapidly dividing lymphocyte progenitors in the bone marrow and thymus. While the percentage of total B cells was not perturbed in the bone marrow (Fig. 2G), we noted a clear decrease in cycling early pro/pre B cells, leading to a relative increase of mature recirculating B cells on day 5 (Fig. 2H,I). Along similar lines, we observed a substantial reduction of thymic cellularity and atrophy upon MAD2 over-expression (Fig. 2J). Immature double-positive (DP) thymocytes were most affected, leading to a relative increase in the percentage of CD4$^+$ and CD8$^+$ single-positive and double-negative (DN) cells (Fig. 2K,L). Within the DN fraction, the DN3 stage appeared most affected, leading to a relative increase in DN1 and DN4 on day 5 (Fig. 2M,N). Granulocytes showed a mild increase within the myeloid compartment in the bone marrow, whereas Mac1$^+$Gr1$^-$ monocytes/macrophages decreased over time (Appendix Fig. S1C,D). Consistent with their resting state, mature T and B lymphocytes in the spleen (Appendix Fig. S1E–I) and lymph nodes were largely unchanged (Appendix Fig. S1J–L). Erythroblasts mildly decreased in the bone marrow but stayed unaffected in the spleen (Appendix Fig. S1M–O).

Together, our data suggest that chronic SAC activation interferes with hematopoiesis by causing the loss of highly proliferative T- and B-cell progenitors. This leads to the mobiliza-tion of HSCPs in the bone marrow and a subsequent increase in CMP and CLPs attempting to replenish hematopoiesis.

## MAD2 overexpressing hematopoietic stem and progenitor cells show reduced reconstitution potential

To study the impact of MAD2 overexpression on the ability of the hematopoietic stem and progenitor cell pool (HSPCs) to salvage the loss of developing leukocytes, we isolated the bone marrow from MAD2-transgenic animals (MR). We tested its colony-forming potential in methylcellulose in the presence or absence of Dox. IL-7 was added to promote pre B cell colony-forming units (CFU-B), M-CSF for macrophage (CFU-M) and G-CSF for granulocyte (CFU-G) colony formation, respectively. We observed that pre B and myeloid colony formation was strongly reduced upon Dox treatment (Fig. 3A). At the same time, untreated *Mad2*-transgenic bone marrow could form colonies at frequencies similar to wild

bone marrow (Fig. 3A). Thus, MAD2 overexpression limits the clonogenic potential of freshly isolated hematopoietic stem and progenitor cells in vitro.

We next investigated the consequences of aberrant MAD2 expression on the hematopoietic system in vivo in a competitive reconstitution setting to avoid potentially detrimental bone marrow failure. Thus, we isolated isogenic *Mad2*-transgenic (MR, Ly5.2$^+$) and *rtTA* (R, Ly5.1/2$^+$) bone marrow (Appendix Fig. S2A) and injected a 50:50 mix into lethally irradiated recipient (Ly5.1$^+$) mice. Bone marrow chimaeras were fed Dox-containing food during hematopoietic reconstitution. The animals showed stable body weight throughout the 12-week observation period (Fig. 3B). Analysis of peripheral blood showed rapid displacement of the *Mad2*-transgenic (MR) Ly5.2$^+$ cells by the *rtTA* single-transgenic (R) Ly5.1/2$^+$ leukocytes. B cells appeared to be most affected, followed by T cells that decreased over time, while myeloid cells were more resilient (Fig. 3C). These results suggest varying susceptibility of different leukocyte subtypes or their respective progenitors to SAC perturbation. In line with our peripheral blood cell analysis, bone marrow, spleen, lymph nodes, and thymus contained mostly hematopoietic cells originating from *rtTA* bone marrow (R) when analyzed 12 weeks after reconstitution (Fig. 3D; Appendix Fig. S2B). Flow cytometry analysis documented reduced percentages of LSK bone marrow cells, a population of cells enriched in HSCPs, and Lin$^-$cKit$^+$ (LK) bone marrow cells, containing committed hematopoietic progenitors. MR-derived myeloid cells and mature T and B cells were also outnumbered (Fig. 3E; Appendix Fig. S2C,D). Together, these findings document reduced fitness of MAD2 overexpressing hematopoietic stem and progenitor cells in vivo, likely leading to a reduced output of leukocytes. However, the impact of chronic SAC activation on leukocyte homeostasis remained uncertain.

## MAD2 overexpression impairs leukocyte homeostasis but fails to induce malignancies

Next, we investigated the consequences of SAC perturbation on hematopoiesis in the presence or absence of a potent cell death inhibitor, i.e., transgenic BCL2. We generated pure bone marrow chimeras using RG, MRG, RG2, and MRG2 animals as bone marrow donors. After reconstitution for 12 weeks, animals were set on a Dox-containing diet to induce *Mad2* transgene expression that was monitored using GFP induction as a surrogate marker

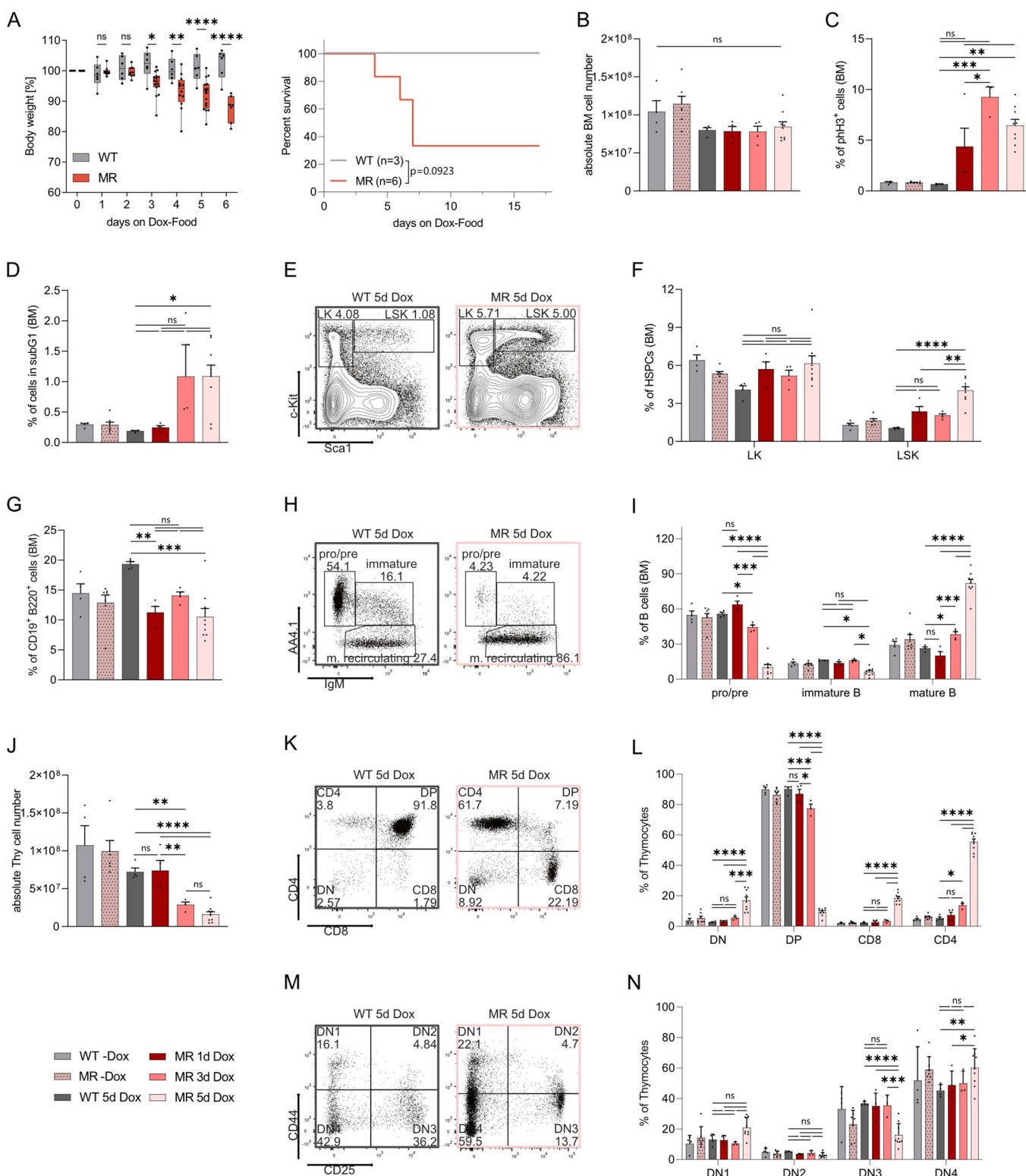

(Fig. 4A). Similar to the 50:50 reconstitution setting, the expression of Ly5.2+ cells allowed us to discriminate between donor (Ly5.2+) and recipient (Ly5.1+) cells. Peripheral blood analysis before and after feeding Dox-containing food confirmed the successful reconstitution of mice with >90% of Ly5.2+ donor cells of various genotypes (Appendix Fig. S3A). This phenotype remained consistent over time with comparable percentages of white (WBC) and red blood cell counts (RBC) in RG and MRG recipients, as well as normal hematocrit (HCT) values. As expected, the presence of the BCL2 transgene initially increased white blood cell counts (Domen

◄   **Figure 2.  Systemic MAD2 overexpression causes myelosuppression and wasting syndrome.**

(A) Body weight and Kaplan–Meier analysis of WT (R) and Mad2-transgenic animals (MR) treated with Doxycycline; R ($n = 3$) MR ($n = 6$), median survival for MR: 7 days. Body weight: WT $n = 6$, MR $n = 7$–16. (B) Bone marrow (BM) cellularity of R and MR mice (both tibiae and femora) on Dox food for the indicated time points. (C) Flow cytometric analysis of mitotic phH3$^+$ bone marrow cells. (D) Quantification of apoptotic BM cells by sub-G1 analysis. (E) Gating strategy used in flow cytometric analyses of HSPCs quantified in (F). (F) Percentage of LK (Lin$^-$ckit$^+$Sca1$^-$) and LSK (Lin$^-$ckit$^+$Sca1$^-$) in BM. (G) BM suspensions stained for total B (CD19$^+$B220$^+$) cells. (H) Gating strategy used in flow cytometric analyses of B cell subsets quantified in (I). (I) BM stained for pro/pre (AA4.1$^+$IgM$^-$), immature (AA4.1$^+$IgM$^+$) and mature recirculating (AA4.1$^-$IgM$^+$) B cells. (J) Thymic (Thy) cellularity of R and MR mice on Dox food for indicated time points. (K) Gating strategy used in flow cytometric analyses of thymocytes quantified in (L). (L) Thymocyte subsets assessed by CD4 and CD8 cell surface marker staining: DN (CD4$^-$CD8$^-$), DP (CD4$^+$CD8$^+$), single-positive (CD4$^+$CD8$^-$), or (CD4$^-$CD8$^+$) cells. (M) Gating strategy used in flow cytometric analyses of thymocytes quantified in (N). (N) Thymocyte subsets assessed by CD44 and CD25 cell surface marker staining of DN thymocytes: DN1 (CD44$^+$CD24$^-$), DN2 (CD44$^+$CD24$^+$), DN3 (CD44$^-$CD24$^+$) and DN4 (CD44$^-$CD24$^-$). (B–N) WT-Dox ($n = 4$), MR-Dox ($n = 7$), WT 5d Dox ($n = 4$), MR 1d Dox ($n = 4$), MR 3d Dox ($n = 3$–4), MR 5d Dox ($n = 9$). Data information: (A–N) Data are shown as mean ± SEM with median and IQR: WT d1 76.5, WT d2 36.5, WT d3 53, WT d4 38, WT d5 37, WT d6 70, MR d1 88.5, MR d2 90, MR d3 88, MR d4 83.5, MR d5 84, MR d6 14. (A) Two-way ANOVA, Šídák's multiple comparisons; Survival curve: Log-rank (Mantel–Cox) test. (B–D, G, J) One-way ANOVA, Tukey's multiple comparisons. (F, I, L, N) Two-way ANOVA, Tukey's multiple comparisons. *$P \leq 0.05$, **$P \leq 0.01$, ***$P \leq 0.001$, ****$P \leq 0.0001$.

et al, 1998; Ogilvy et al, 1999) in RG2 and MRG2 recipients. Still, RBC and HCT were comparable to that seen in RG and MRG reconstituted animals (Appendix Fig. S3B). The distribution of B and T lymphocytes and myeloid cells in the peripheral blood did not change significantly during Dox treatment (Appendix Fig. S3C).

Notably, the fraction of GFP$^+$ cells in the peripheral blood increased already on day 1 after providing Dox-containing food. About 60% of GFP$^+$ cells were observed after 28 days in RG and RG2 recipients (Fig. 4B), while animals reconstituted with *Mad2*-transgenic bone marrow (MRG) showed a significantly lower percentage of GFP$^+$ blood cells than controls (RG). Curiously, while MRG2 recipients showed a faster increase in the percentage of GFP$^+$ cells, as seen on day 10, potentially due to impaired cell death, both MRG and MRG2 recipients seemingly reached a plateau of about 20% and 30% GFP$^+$ cells, respectively (Fig. 4B), but this difference was not statistically significant. This indicated reduced fitness or impaired proliferative capacity of MAD2 overexpressing blood cells aiming to replace resident leukocytes, even when BCL2 was overexpressed.

After 10 weeks of Dox treatment, mice were sacrificed for organ analysis. BCL2 and GFP-transgene expression were confirmed along with HA-MAD2 in the bone marrow of recipient mice by immunoblot (Fig. 4C). The relative abundance of T, B, and myeloid cells was broadly comparable across the different genotypes and organs, showing similar percentages of these populations in flow cytometric analyses. A notable exception was an increase in myeloid cell infiltrates into lymph nodes in response to MAD2 overexpression, suggesting sterile inflammation (Appendix Fig. S3D–F). Most importantly, we observed that MAD2 expression from the *Col1a* locus was highly variegated in different hematopoietic organs. The highest percentages of GFP$^+$ cells were found in the thymus (Thy), followed by the bone marrow (BM) and the spleen (SP) (Fig. 4D–F).

On the one hand, organ cellularity was similar in bone marrow across genotypes (Fig. 4D). At the same time, BCL2 overexpression led to the expected increase in splenocytes (Domen et al, 1998; Ogilvy et al, 1999) in the absence or presence of MAD2 (Fig. 4E). On the other hand, MAD2 overexpression led to a selective reduction in thymic cellularity (Fig. 4F), caused by a significant drop in CD4$^+$CD8$^+$ immature double-positive cells that was not seen on a BCL2 transgenic background (Fig. 4G). BCL2 overexpression also led to a general redistribution of different

thymocyte subsets (Ogilvy et al, 1999), but this phenomenon was not affected by excess MAD2 (Fig. 4G). Evaluating the leukocyte subset distribution within the GFP$^+$ compartment, we noted that MAD2 overexpression led to a clear reduction of reporter-positive leukocytes. This reduction within bone marrow spleen and lymph nodes (LN) comparing MRG with RG recipients was buffered by BCL2 overexpression (MRG2 vs. RG2) (Fig. 4H–J).

*Vav-BCL2* transgenic animals show severely reduced lifespan due to autoimmune disease or follicular lymphoma development, exacerbated in a bone marrow reconstitution setting (Egle et al, 2004; Labi et al, 2014). Therefore, we could not study the long-term consequences of cell death inhibition after MAD2 overexpression in this model. Instead, to monitor the spontaneous transformation potential of MAD2 alone, we followed a small cohort of MRG mice for one year on Dox-containing food. Analysis of peripheral blood for the percentage of total Ly5.2$^+$ cells showed a stable reconstitution with MRG bone marrow (Fig. EV3A). HCT, RBC and WBC in these animals were stable over time, as was the distribution of leukocytes (Fig. EV3B,C). The fraction of GPF$^+$ cells in the blood increased gradually, reaching more than 90% after 12 months (Fig. EV3D). Interestingly, after one month, the percentage of GFP$^+$ myeloid cells reached a plateau of about 70%. Turnover of B and T cells required more time, reaching >80% after the endpoint of our blood cell analysis (Fig. EV3E). A final analysis of primary and secondary hematopoietic organs after 12 months confirmed an increased percentage of phH3$^+$ bone marrow, suggesting the *Mad2* transgene was still expressed and active in hematopoietic cells (Fig. EV3F). Consistently, MAD2 overexpression was well-detectable on protein level in bone marrow, spleen and thymus by western blot (Fig. EV3G–I). Organ cellularity and the composition of hematopoietic cells in bone marrow, spleen, lymph nodes (Fig. EV3J) and thymus (Fig. EV3K) were comparable to that of MRG animals analyzed after 2 months (Fig. 4). None of the animals did develop signs of malignant disease, as monitored by flow cytometric analysis of primary and secondary lymphatic organs using a range of different antibodies (see "Reagent and Tools table"). No expansion of a distinct population of myeloid cells or lymphocytes was observed by flow cytometry, nor did we observe abnormal white blood cell counts, splenomegaly, enlarged thymus or lymph nodes upon sacrifice.

In summary, we noted clear signs of myeloablation in MRG recipients after transgene induction that were ameliorated by BCL2 overexpression. This suggests a role for mitochondrial apoptosis in

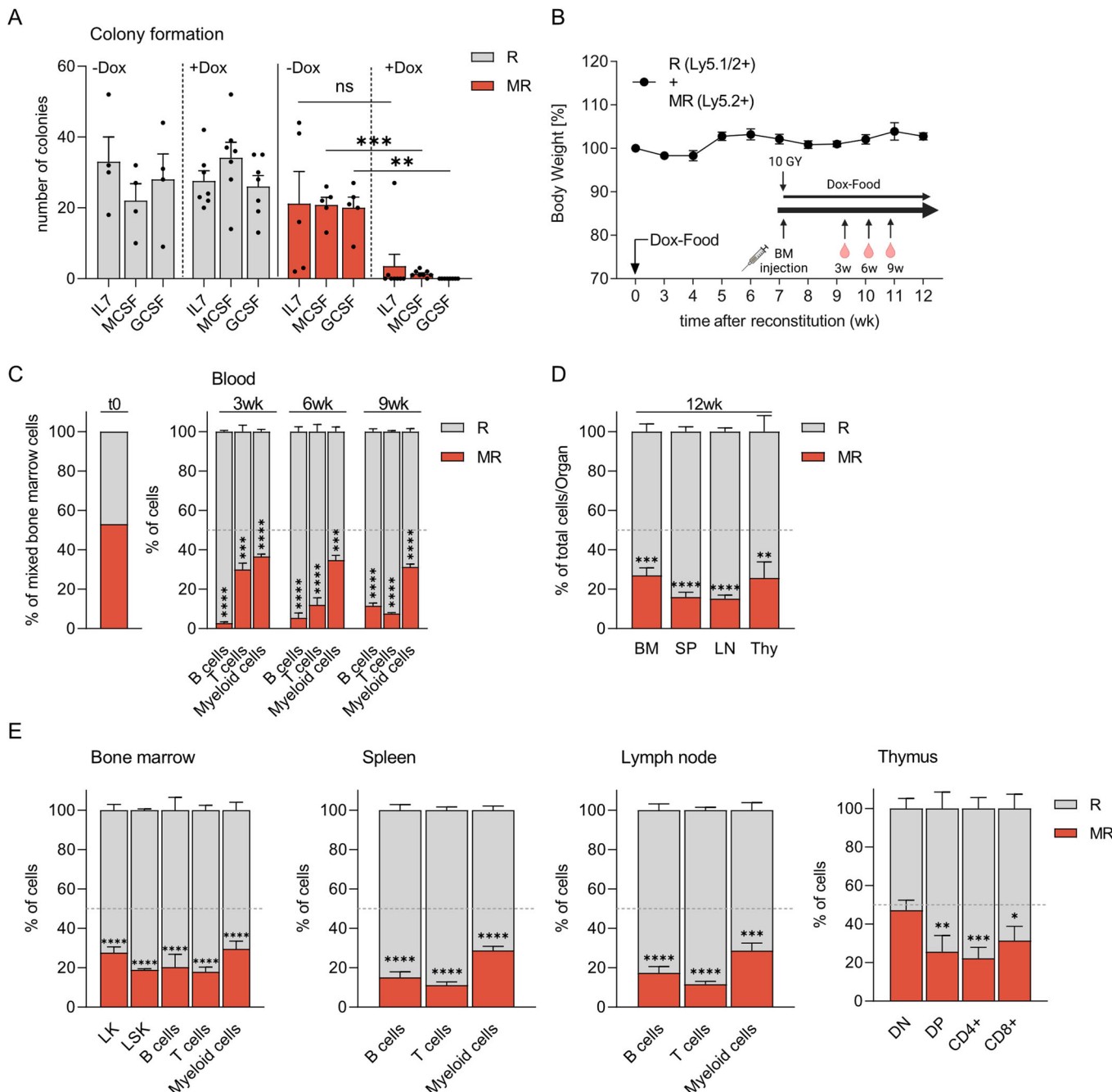

**Figure 3. MAD2 overexpression perturbs fitness of hematopoietic stem and progenitor cells.**

(A) Colony formation potential of MAD2 (MR) and rtTA (R) transgenic bone marrow cells from individual mice in methylcellulose assays in the presence (+) or absence (−) of Doxycycline; R+Dox (n = 7), MR+Dox (n = 8), R-Dox (n = 4), MR-Dox (n = 5). (B) Scheme of bone marrow reconstitution experiments and body weight analysis of chimaeric animals receiving a 50:50 mix of R (Ly5.1/2+) and MR (Ly5.2+) BM cells. (C) The degree of chimaerism of peripheral blood cells was assessed by flow cytometry over time and compared to time zero (t0). Peripheral blood was sampled on indicated time points after reconstitution and diet change to analyze the content of B (CD19+B220+), T (CD4+&CD8+), and myeloid cells (CD11b+); percentages of R (Ly5.1/2+) and MR (Ly5.2+) cells within each cell type are shown. (D) The degree of chimaerism in hematopoietic organs of reconstituted animals was analyzed after 12 weeks by flow cytometric analysis (BM bone marrow, SP spleen, LN lymph node, Thy thymus). (E) BM, SP, LN, and Thy were analyzed after 12 weeks for the percentage of R and MR cells in different hematopoietic cell types: LK (Lin−Sca1−cKit+) and LSK (Lin−Sca1+cKit+), B (CD19+B220+), T (CD4+ and CD8+), myeloid cells (CD11b+), DN (CD4−CD8−), DP (CD4+CD8+), T helper (CD4+), and T cytotoxic (CD8+) cells. (B–E) n = 4 animals reconstituted with 50:50 mix. Data information: (A–E) Data are shown as mean ± SEM. (A, D, E) Unpaired t test, Welch's correction. (C) Two-way ANOVA, Šídák's multiple comparisons. *P ≤ 0.05, **P ≤ 0.01, ***P ≤ 0.001, ****P ≤ 0.0001.

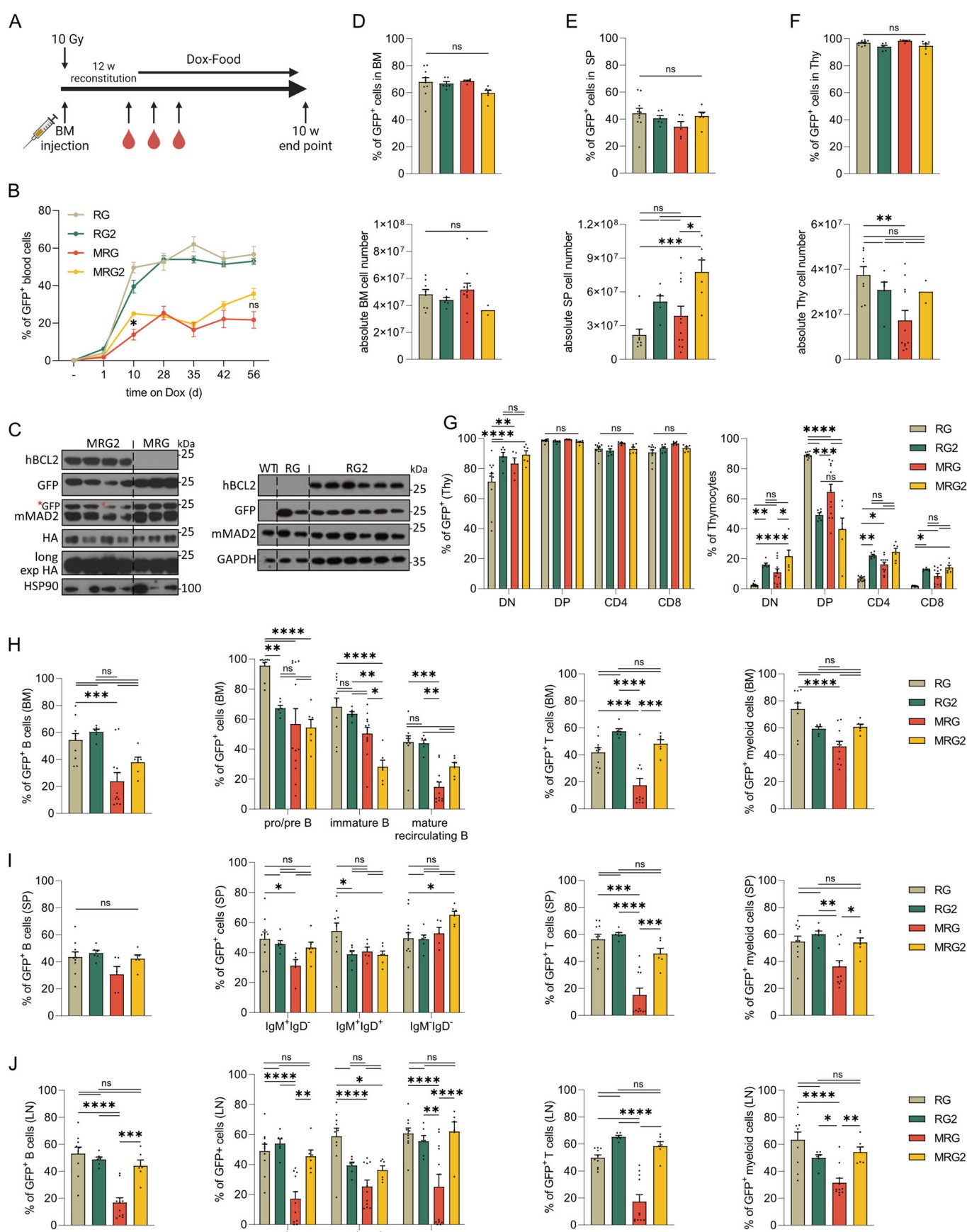

**Figure 4. MAD2-induced myelosuppression is ameliorated by transgenic *BCL2*.**

(A) Scheme of bone marrow reconstitution analysis. Twelve weeks after successful bone marrow reconstitution mice received Dox-containing food. Peripheral blood was sampled before and after diet change. After additional 10 weeks mice were sacrificed for further analysis. (B) Total GFP expression in the peripheral blood of chimeras of indicated genotypes was monitored over time. (C) The bone marrow of reconstituted animals was subjected to western blot analysis using the indicated antibodies (*: persisting GFP signal after re-probing, WT: non-reconstituted mouse). (D–F) Flow cytometric analysis of GFP expression and quantification of absolute cell counts in (E) bone marrow (BM, both femora and tibiae), (E) spleen (SP), and (F) thymus (Thy) of mice of the indicated genotypes 12 weeks after reconstitution and additional 10 weeks of Dox treatment. (G) Analysis of GFP expression of thymocyte subsets quantified in (H). (H) Thymocytes were stained using CD4 and CD8-specific antibodies to identify CD4$^-$CD8$^-$ double-negative (DN), CD4$^+$CD8$^+$ double-positive (DP), CD4$^+$CD8$^-$ single-positive and CD4$^-$CD8$^+$ thymocytes. (H–J) GFP$^+$ cells within (H) BM, (I) SP, and (J) LN (lymph node) were analyzed for the presence of total B cells (CD19$^+$B220$^+$) and different B cell subsets: pro/pre (AA4.1$^+$IgM$^-$), immature (AA4.1$^+$IgM$^+$) and mature recirculating (AA4.1$^-$IgM$^+$), or IgM$^+$IgD$^-$, IgM$^+$IgD$^+$, IgM$^-$IgD$^-$ B cells of total B cells; T (CD4$^+$ and CD8$^+$) and myeloid cells (CD11b$^+$). (B–J) RG ($n = 8$–10), RG2 ($n = 6$), MRG ($n = 5$–12), MRG2 ($n = 2$–6). Data information: (A–J) Data are shown as mean ± SEM. (B) Mixed-effect model, Šidák multiple comparisons. (D–F, H–J) One-way ANOVA, Tukey's multiple comparisons. (G–J) Two-way ANOVA, Tukey's multiple comparisons. *$P \leq 0.05$, **$P \leq 0.01$, ***$P \leq 0.001$, ****$P \leq 0.0001$.

response to chronic SAC perturbation in white blood cells. Moreover, due to variegated transgene expression, MAD2 over-expression in the hematopoietic system is tolerated with minimal impact on organ cellularity. Surprisingly, MAD2 overexpression per se appears insufficient to promote spontaneous blood cancer within 12 months, likely due to increased apoptotic priming.

## Chronic SAC perturbation causes severe colitis and gastrointestinal syndrome

Our results in reconstitution experiments suggested that myelo-suppression is limited and *Mad2* transgene expression variegated in blood cells, hence not causal for the rapid wasting of MR animals. Based on the similarities of the phenotype seen with radiation disease, we interrogated the gastrointestinal (GI) tract. GI-tract integrity and barrier function rely on the constant and rapid replacement of short-lived gastrointestinal epithelial cells, shed off at the tip of villi and replaced by newly differentiating cells originating from the crypt-based stem cell and the transient amplifying pool (Barker et al, 2007; Blander, 2016).

Histopathological assessment of the intestine revealed a severe colitis/enteritis phenotype in *Mad2*-transgenic animals (Fig. 5A,B), likely explaining the animals' weight loss and rapid decline. MAD2 transgene expression was verified in the GI tract after Dox addition by IHC (Fig. 5C) and western blot of total intestinal lysates (Fig. EV4A). We also confirmed increased phH3$^+$ cells in the colonic and small intestinal crypts of *Mad2*-transgenic animals, indicating the expected mitotic arrest and SAC activation (Fig. 5C,D). This correlated well with increased active cleaved Caspase-3$^+$ cells in both the colon and the small intestine, documenting increased cell death (Fig. 5C,E) and elevated p21 levels (Figs. 5C and EV4A). Evaluating expression levels of different BCL2-family proteins indicated lower levels of BCL2 and BIM, a reported APC substrate (Fava et al, 2013; Wan et al, 2014), in response to Dox treatment, while levels of BCLX and MCL1 appeared unchanged (Fig. EV4A).

We hypothesized that the deletion of pro-apoptotic BIM together with NOXA, both needed for mitotic cell death in epithelial cancer cells (Haschka et al, 2015; Sloss et al, 2016), might ameliorate the wasting syndrome noted in *Mad2*-transgenic mice by allowing GI-tract stem cells or transient amplifying cells experiencing mitotic delays to evade cell death. Indeed, we could observe that the loss of *Bim* and *Noxa* damped the colitis phenotype (Fig. 5F) and extended the lifespan of *Mad2*-transgenic (MRBN) mice (Figs. 5G and EV4B). Wondering whether the

deletion of both BIM and NOXA is necessary for this survival advantage, we also analyzed MAD2 overexpressing animals lacking only *Noxa* (MRN) or *Bim* (MRB). Of note, MRN mice developed severe colitis followed by a wasting syndrome similar to MAD2 overexpressing MR mice, while MRB animals showed no pathological phenotype (Fig. 5F,G). Surprisingly, we observed a reduction in the number of phH3$^+$ cells in the absence of *Bim* in MRBN and MRB mice (Fig. 5H) despite strong HA expression in the intestine (Fig. EV4B). This suggests that these cells may escape mitotic arrest and death in mitosis. Importantly, the co-deletion of *Bid* and *Puma/Bbc3* (MRBP), two other pro-apoptotic BH3-only proteins, could not rescue the colitis/enteritis phenotype (Fig. 5F) nor extend the lifespan upon aberrant MAD2 expression (Fig. 5G). Of note, one of the two reported isoforms of the BH3-only protein BMF (Grespi et al, 2010) was found mildly increased upon MAD2 overexpression by western blot analysis (Appendix Fig. S4A). Taken together, our data suggest that delays in mitotic cell death caused by loss of BIM can overcome the detrimental effects of impaired SAC proficiency in the GI tract, allowing escape from mitotic arrest. If other BH3-only proteins not tested here, such as BMF, contribute to cell death initiation in the GI-tract remains to be explored.

## Loss of BH3-only proteins fails to prevent MAD2-induced myelosuppression

To evaluate the contribution of mitotic cell death to the observed myelosuppression, we also investigated the impact of BH3-only protein deletion on blood cell formation and survival. Animals were analyzed once they became moribund or on day 17 when remaining unaffected by Dox treatment (Fig. 5G).

Similar to our observations in the GI tract, loss of *Bim* caused a decrease in the fraction of mitotic phH3$^+$ bone marrow cells after Dox treatment in MRB and MRBN mice (Fig. 6A). Bone marrow cellularity and the percentage of HSPCs were not significantly changed within the treatment window (Fig. 6B,C). However, loss of *Bim* and *Noxa* did not translate into rescuing progenitor B cells in the bone marrow (Fig. 6C) or T-cell progenitors in the thymus (Figs. 6D and EV4D).

Splenic cellularity and weight were significantly increased in MRBN and MRB animals (Figs. 6E and EV4E). However, this can be explained by the splenomegaly caused by the loss of *Bim* (Bouillet et al, 1999). No changes were observed in the distribution of IgM$^+$D$^+$ mature B cells, CD4$^+$ or CD8$^+$ T cells, as well as myeloid cells in the spleen (Fig. 6E–G) or lymph nodes (Figs. 6H

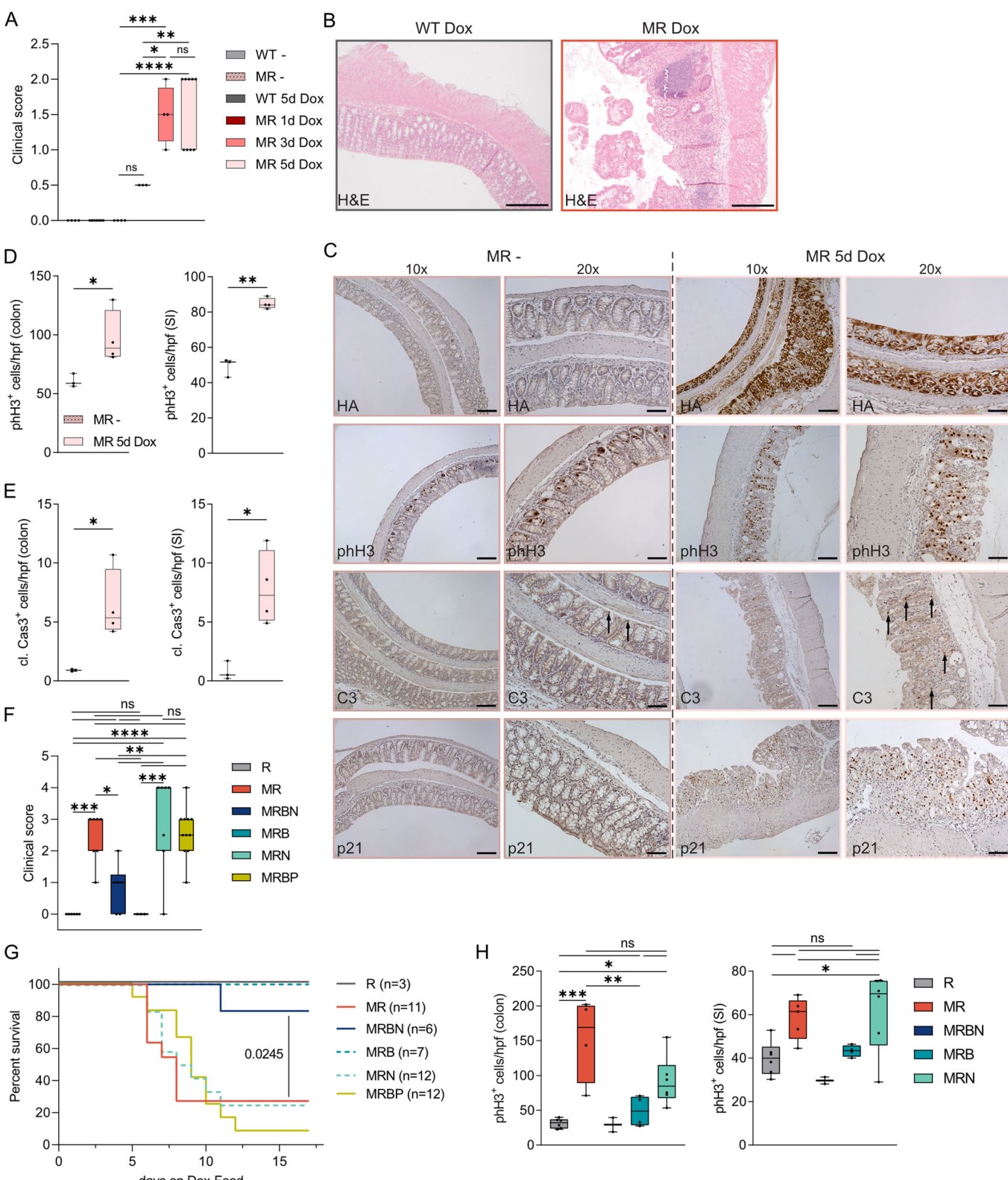

and EV4F,G). These findings contrast observations made in BCL2 overexpressing HoxB8 cells (Fig. 1) and mice reconstituted with BCL2 transgenic bone marrow, where some degree of protection from the consequences of MAD2 overexpression was noted (Fig. 3).

In line with the above, the colony formation potential of *Mad2*-transgenic HSPCs was not restored by combined loss of BIM and NOXA, nor by loss of BID and PUMA, as tested in methylcellulose assays (Fig. EV5A). Moreover, the loss of both BH3-only proteins

Figure 5. Systemic MAD2 overexpression causes in the gastrointestinal atrophy ameliorated by loss of *Bim/Bcl2l11*.

(A) Clinical score of the intestine of animals on regular diet or Dox-containing food. WT ($n=3$), MR ($n=7$), WT 5d Dox ($n=3$), MR 1d Dox ($n=3$), MR 3d Dox ($n=4$), MR 5d Dox ($n=9$). (B) H&E staining of paraffin-embedded sections from Swiss rolls generated from healthy and terminally sick animals. Scale bar 200 µm. (C) Representative pictures of MR animals on regular diet (MR-) and 5d on Dox food (MR 5d Dox), stained for HA, phH3, cleaved Caspase-3 (C3) and p21 (×10 and ×20 magnification are shown, scale bar 1 mm). (D, E) Quantification in colon and small intestine (SI) stained for (D) mitotic cells (phH3$^+$) and (E) cleaved C3$^+$ cells. MR- ($n=3$), MR 5d Dox ($n=4$); randomly selected mice. (F) Clinical score of animals with the indicated genotypes at the time of sacrifice (>17% loss of body weight) or after 17 days. R ($n=6$), MR ($n=7$), MRBN ($n=6$), MRB ($n=3$), MRN ($n=7$), MRBP ($n=11$). (G) Kaplan–Meier analysis of mice overexpressing Mad2 in the absence or presence of different BH3-only proteins. (H) Quantification of colon and small intestine (SI) stained for mitotic cells (phH3$^+$). R ($n=6$), MR ($n=4$), MRBN ($n=2$), MRB ($n=4$), MRN ($n=6$). Data information: (A, D–F, H) Data shown as Min to Max with median and (A) IQR: WT- 0, MR- 0, WT 5d Dox 0, MR 1d Dox 0, Mad2 3d Dox 0, Mad2 3d Dox 0.5, Mad2 5d Dox 1. (D) MR- unknown Q2 58.8, MR 5d Dox 29.3. (E) MR- unknown Q2 0.9, MR 5d Dox 3.7. (F) R 0, MR, MRBN 1, MRB 0, MRN 2, MRBP 1. (H, colon) R 11.7, MR 91.25, MRBN unknown Q2 29.25, MRB 37.85, MRN 29.17. (H, SI) R 9.1, MR 17.6, MRBN unknown Q2 29.7, MRB 3.7, MRN 24. (D, E) Unpaired *t* test, Welch's correction. (A, F, H) One-way ANOVA, Tukey's multiple comparisons. (G) Survival curve: Log-rank (Mantel–Cox) test, median survival MR: 8 days, MRBP: 9 days, MRN: 8.5 days. *$P \leq 0.05$, **$P \leq 0.01$, ***$P \leq 0.001$, ****$P \leq 0.0001$.

caused only a very modest delay in mitotic cell death of HoxB8-immortalized progenitors after MAD2 overexpression ex vivo, contrasting findings made with BCL2 overexpression (Figs. 1, EV1E, and EV5B–D). Moreover, we failed to see significant protection from apoptosis in progenitor cells lacking both BH3-only proteins, BID and PUMA (Fig. EV5D). The BH3-only protein BMF, mentioned above, was not detectable in HoxB8-immortalized progenitor cells, neither in the steady state nor upon MAD2 overexpression (Appendix Fig. S4B). This suggests that apoptosis effectors next or alternative to BIM must be at play to remove hematopoietic cells experiencing substantial mitotic delays in vivo.

## Discussion

Defects in the spindle assembly checkpoint are linked to developmental defects, premature aging phenotypes, and malignant disease while targeting this checkpoint has been exploited to treat cancer patients for several decades (Marzo and Naval, 2013; Trakala et al, 2021; Yamada and Gorbsky, 2006). Our understanding of systemic and cellular responses contributing to human pathology, cancer treatment efficacy, and side effects is still incomplete. MAD2 overexpression delays mitosis, fosters chromosome miss-segregation and aneuploidy in fibroblasts, and accelerates MYC-driven and spontaneous tumor initiation in transgenic mice (Sotillo et al, 2007). Of note, Sotillo et al (2007) showed that the tumor spectrum observed in MAD2 overexpressing mice phenocopied those found in human cancer with high MAD2 levels and poor prognosis, such as hepatocellular and lung carcinomas, as well as diffuse large B-cell lymphomas (DLBCL). However, the need for additional genomic alterations to allow tumor formation might explain the rather long tumor latency reported in this model (Simonetti et al, 2019; Sotillo et al, 2007). Remarkably, anti-tumorigenic effects of MAD2 overexpression have also been noted, e.g., in a Human Epidermal Growth Factor Receptor 2 (*Her2*)-driven breast cancer model. Pro-tumorigenic vs. anti-tumorgenic effects may be linked to transgene expression levels, where high levels are becoming incompatible with cell survival in some tissues. Notably, exogenous MAD2 expression in mammospheres isolated from *Col1a-HA-MAD2* mice were even higher than those found in the originally used *Tet-O-HA-MAD2* strain (Rowald et al, 2016). This suggests that the differential effects of MAD2 overexpression on cancer can be tissue or cell-type-

dependent and depend on apoptotic priming, facilitating or restricting transformation.

Consistent with the latter, we observed increased cell death in vitro in myeloid progenitor cells overexpressing MAD2 (Figs. 1 and EV1). In line with this finding, we failed to note an increase in poly- or aneuploidy in HoxB8 model cell lines. Moreover, even though we could delay cell death by BCL2 overexpression, we did not detect increased levels of aneuploidy by scWGS (Fig. EV2). How chronic MCC signaling is overcome in these cells remains to be investigated. Increased levels or activity of p31comet may be involved (Díaz-Martínez et al, 2014; Varetti et al, 2011), but lack of commercially available mouse-specific antibodies precluded us form investigating this possibility. Of note, cell death was still seen at later time points, suggesting secondary necrosis or the activation of alternative cell death pathways during ex vivo culture of BCL2 transgenic cells (Fig. 1). Consistently, we also observed that MAD2 overexpression causes the loss of cycling lymphocyte progenitors (Fig. 2) and limits the clonogenic potential of primary hemato-poietic stem and progenitor cells (HSCP) in vitro and in vivo, best seen in competitive reconstitution assays (Fig. 3). If this happened solely by the induction of apoptosis, as seen in our ex vivo culture, or whether the induction of cell cycle arrest and senescence may contribute could not be distinguished using these readouts. The use of fully reconstituted animals expressing MAD2 together with a BCL2 transgene, however, supports a role of cell death induction in leukocytes also in vivo, as consequences of MAD2 overexpression were ameliorated by blocking mitochondrial apoptosis (Fig. 4). In light of these observations, the lack of spontaneous blood cancer development in animals overexpressing MAD2 in the hematopoie-tic system may not come as a surprise (Figs. 4 and EV3). However, this contrasts findings originally made in *Mad2*-transgenic animals that occasionally developed lymphomas during their first year of life and hence survived transgene expression long term (Sotillo et al, 2007). One may speculate about differences in genetic background and the use of a different transactivator for transgene expression, i.e., *CMV-rtTA* vs. *R26rtTA*, used here, that are known to differ in strength and cell-type-specificity in the hematopoietic system (Takiguchi et al, 2013). Moreover, surviving cells with complex karyotypes might be cleared more effectively by NK cells in our model. The latter phenomenon was recently reported in cell line studies describing the killing of polyploid and aneuploid cells by NK cells ex vivo (Garcia-Carpio et al, 2023; Santaguida et al, 2017; Wang et al, 2021).

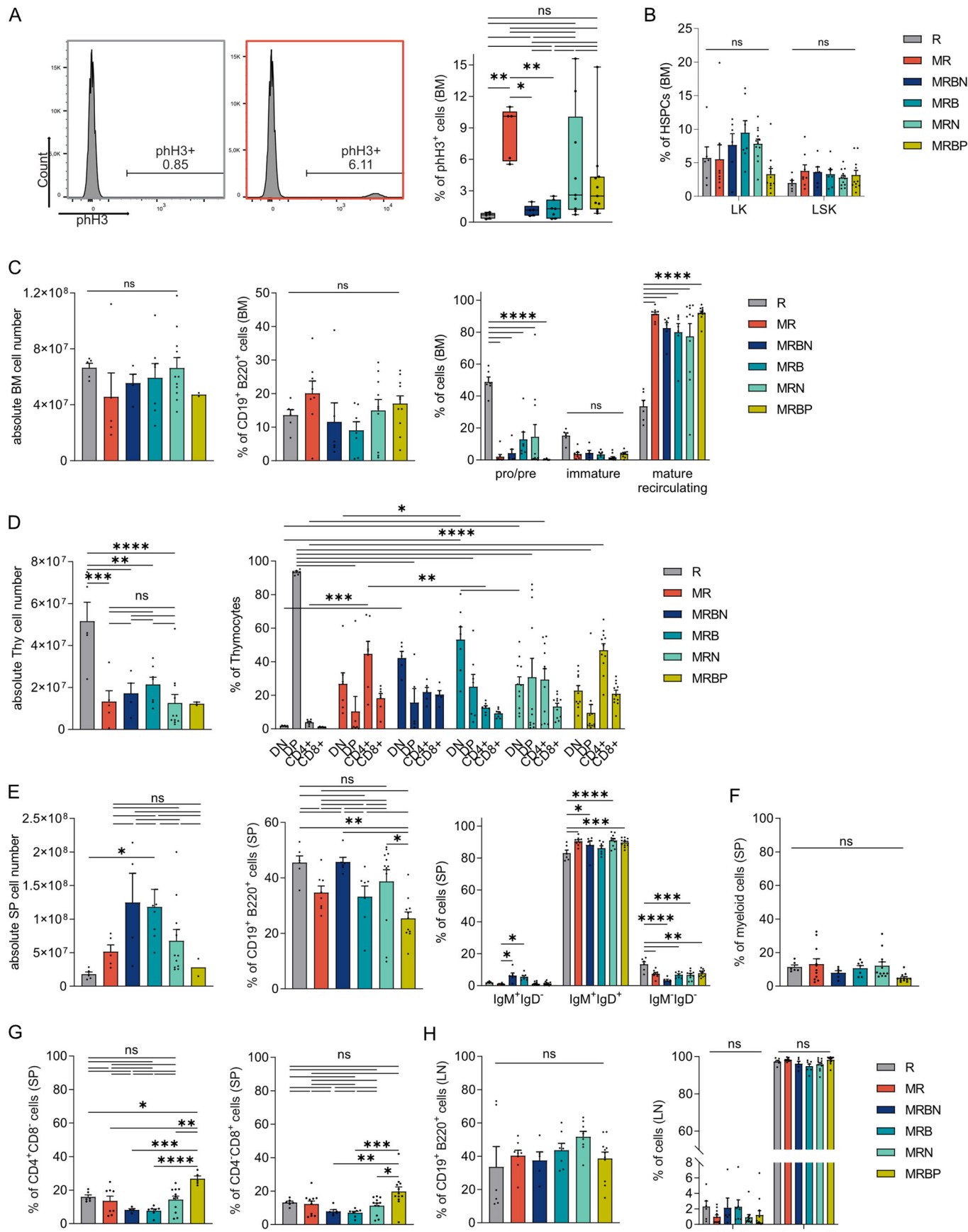

**Figure 6. MAD2-induced myelosuppression is not ameliorated by loss of BIM and NOXA.**

(A, B) Bone marrow (BM) of WT, MR, MRBN ($Mad2/rtTA/Bim^{-/-}/Noxa^{-/-}$), MRB ($Mad2/rtTA/Bim^{-/-}$), MRN ($Mad2/rtTA/Noxa^{-/-}$) and MRBP ($Mad2/rtTA/Bid^{-/-}/Puma^{-/-}$) mice when terminally sick or at the end time point of 17 days on Dox food stained for (A) mitotic (phH3$^+$, gating strategy shown in flow cytometry shown) and (B) LK (Lin$^-$ckit$^+$Sca1$^-$) and LSK (Lin$^-$ckit$^+$Sca1$^-$) cells. (C) BM cellularity (both femora and tibiae) and staining for B (CD19$^+$B220$^+$) cells and subsets: pro/pre (AA4.1$^+$IgM$^-$), immature (AA4.1$^+$IgM$^+$) and mature recirculating (AA4.1$^-$IgM$^+$) B cells. (D) Thymic (Thy) cellularity and thymocyte subsets identified by immune-staining: DN (CD4$^-$CD8$^-$), DP (CD4$^+$CD8$^+$), single-positive T helper (CD4$^+$CD8$^-$) and T cytotoxic (CD4$^-$CD8$^+$) cells. (E–G) (E) Spleen (SP) cellularity and staining for B (CD19$^+$B220$^+$) cells defining IgM$^+$IgD$^-$, IgM$^+$IgD$^+$, IgM$^-$IgD$^-$ subsets, (F) myeloid (CD11b$^+$) cells, and (G) T cells (CD4$^+$CD8$^-$, CD4$^-$CD8$^+$). (H) Lymph node (LN) stained for B (CD19$^+$B220$^+$) cells, germinal center (GC, CD38$^{int}$CD59$^+$) and naive (CD38$^{int}$CD59$^-$) B cells. R ($n = 6$), MR ($n = 5$–8), MRBN ($n = 4$–6), MRB ($n = 7$) and MRN ($n = 10$), MRBP ($n = 2$–11). Data information: (A) Data shown as Min to Max with median and IQR: R 0.48, MR 4.73, MRBN 0.92, MRB 1.8, MRN 8.88, MRBP 3.12. (B–H) Data are shown as mean ± SEM. (A–H) One-way or two-way ANOVA, Tukey's multiple comparisons. *$P \leq 0.05$, **$P \leq 0.01$, ***$P \leq 0.001$, ****$P \leq 0.0001$.

Moreover, alterations in the non-hematopoietic compartment due to *Mad2* transgene expression may add to the differences in tumor incidence. Consistent with this idea, acceleration and delays in tumor development were shown upon MAD2 overexpression. In a KRAS-driven mouse model of lung cancer, MAD2-induced CIN accelerated tumourigenesis and relapse after oncogene silencing (Sotillo et al, 2010). Conversely, in a *Her2*-driven mouse model of breast cancer, MAD2 overexpression caused a delay in tumor onset but facilitated tumor persistence after oncogene withdrawal (Rowald et al, 2016). Cell-type-specific differences in response to SAC impairment, driving transformation in combination with different oncogenic drivers may likely explain these results.

Based on our reconstitution experiments using GFP as a reporter for MAD2 expression, we concluded that reduced fitness of MAD2 overexpressing HSCs and initial myeloid aplasia was overcome by transgene-negative blood cells due to variegated transgene expression (Fig. 4), allowing the long-term survival of these mice. This phenomenon was apparently not pronounced in the GI-tract stem cells, where transgene overexpression was detectable throughout all crypts in the small intestine and colon over extended periods with no signs of variegated expression patterns (Figs. 5 and EV4). Consistently, gastrointestinal stem cells and the transient amplifying pool seemed to be the most affected cells by MAD2 overexpression and extended mitosis as an imbalance in MAD2 expression in either direction can cause prolonged SAC activation (Michel et al, 2001; Sotillo et al, 2007). Our finding is well in line with a recent report showing that timed conditional deletion of MAD2 in *Mad2l1^{f/f}* mice carrying a *CRE^{ERT2}* transgene triggers a similar phenotype with rapid weight loss, mitotic abnormalities and increased apoptosis in intestinal crypts, causing rapid animal death (Schukken et al, 2021).

Our study shows myelosuppression due to MAD2 overexpression is mainly limited to early cycling T and B lymphocyte progenitors. Accordingly, primary hematopoietic organs, bone marrow and the thymus were affected most strongly, with thymocytes and early pro/pre B-cell progenitors severely reduced (Figs. 2 and 3). Moreover, we found an increase in the fraction of HSPCs in the bone marrow of *Mad2*-transgenic mice (Fig. 2), which indicates a mobilization response to compensate for the loss of T and B-cell lymphoid progenitors. Consistently, we noted an accumulation of CMP and CLPs in the bone marrow (Appendix Fig. S1). Alternatively, a block in differentiation may lead to an accumulation of HSPCs, fostering myelosuppression. Yet, based on all our findings, the former appears more compatible with the cell death-inducing effects of MAD2 overexpression in the HoxB8 model and the drop in viability of early B- and T-cell progenitors.

In contrast, the percentage and number of mature T and B cells appeared normal in the reconstitution setting. Still, both were found to be reduced within the GFP$^+$ fraction of cells (Fig. 4), mimicking results from mixed bone marrow chimeras (Fig. 3). As those cells are already differentiated and do not proliferate unless challenged by antigens, the noted drop may reflect a loss of influx of new cells rather than deletion of mature lymphocytes. Yet, BCL2 overexpression ameliorated the loss of immature and mature lymphocytes within the reconstitution setting (Fig. 4), suggesting that some surviving progenitors may be able to mature further in the presence of transgenic MAD2.

It is worth mentioning that mice that survived the systemic overexpression of MAD2 did not show such signs of myelosuppression at the time of analysis but were still diagnosed with colitis of lower grades. This suggests intestinal stem cells are more vulnerable to MAD2 overexpression and unable to escape transgene-induced effects unless cell death is perturbed. Our most striking finding is that we could increase the lifespan of *Mad2*-transgenic animals by the co-deletion of the pro-apoptotic protein BIM and NOXA. Our follow-up experiments identified BIM as the most critical cell death effector, as the loss of several other BH3-only proteins tested, including the combination of BID and PUMA or NOXA alone, provided no noticeable survival advantage. This finding complements our previous in vitro studies, where we could show that upon extended mitotic arrest induced with microtubule-targeting agents, anti-apoptotic MCL1 and NOXA are co-degraded, thereby releasing BIM that can initiate apoptosis (Haschka et al, 2015). BIM is a potent pro-apoptotic protein, and it was shown that the deletion of *Bcl2l11* in mice causes lymphadenopathy but can also restore the detrimental effects of loss of *Bcl2* on mature B and T cell homeostasis (Bouillet et al, 1999). As such, it was somewhat unexpected to learn that loss of BIM protein expression, alone or in combination with NOXA, failed to dampen myelosuppression and restore early lymphopoiesis in the thymus or bone marrow (Fig. 6). Based on our observations using BCL2 overexpression in cultured cells (Figs. 1 and EV5), we conclude that BIM must act in conjunction with other BH3-only proteins but NOXA. BMF (Pan et al, 2023) or BID (Wang et al, 2014), have both been implicated in mitotic cell death and hence appear to be plausible candidates worth testing, as both were reported to genetically interact with BIM in other contexts (Kaufmann et al, 2009; Labi et al, 2014). Alternatively, developing T and B cell progenitors may activate alternative cell death pathways when facing mitotic delays that do not rely on BIM or any of the combinations we tested. Consistently, cell-type-dependent differences regarding the role of BIM in mitotic cell death have been reported (Haschka et al, 2015; Wan et al, 2014). What stands out, however, is the observation that loss

of BIM correlated with a reduction of phH3$^+$ cells in the GI tract and bone marrow in situ (Figs. 6A and EV4C), contrasting findings in HoxB8 cells (Fig. EV5B). While indicating higher rates of MCC inactivation in both cases, this appears to enhance cell survival in the GI tract, but not the bone marrow. How the loss of BIM would affect adaptation is unclear. Still, similar to p31comet exerting undefined anti-apoptotic roles in mitotically arrested cells (Díaz-Martínez et al, 2014; Varetti et al, 2011; Fava and Villunger, 2014)), BIM may restrain MCC function by yet-to-be-defined mechanisms.

In conclusion, our study identifies BIM as rate-limiting for apoptosis induction in the gastrointestinal epithelium in response to SAC perturbation. In contrast, only BCL2 overexpression, rather than BH3-only protein deficiency, demonstrated the capacity to mitigate myelosuppression. These findings underscore the significance of tissue-specific survival dependencies in response to SAC perturbation. This aspect has potential implications for treatment strategies seeking to merge BH3 mimetics with spindle poisons or innovative inhibitors targeting the cell cycle machinery, such as MPS1, PLK1, or Aurora kinase inhibitors. The varying responses within different tissues may influence the therapeutic effectiveness and the potential side effects of such treatment regimens.

# Methods

### Reagents and tools table

| Reagent/resource | Reference or source | Identifier or catalog number |
| --- | --- | --- |
| **Experimental Models** | | |
| in Methods | | |
| **Recombinant DNA** | | |
| NA | | |
| **Antibodies** | | |
| Mouse-anti-HA | Covanze | clone MMS-101P (1:1000 for WB) |
| Rabbit-anti-Mad2 | Bethyl | clone A300-301A (1:1000 for WB) |
| Rabbit-anti-GFP | Gift from Stephan Gelay | clone SG4-1 (1:1000 for WB) |
| Mouse-anti-c-Mad2 | Santa Cruz | sc-65492. 107-276-3 (1 µg for IP) |
| Rabbit-anti-MCL1 | Rockland | clone 600-401-394 (1:1000 for WB) |
| Mouse-anti-human BCL2 | Gift from Andreas Srasser | clone S100 (1:500 for WB) |
| Mouse-anti-mouseBCL2 | Biolegend | clone BCL/10C4 (1:1000 for WB) |
| Rabbit-anti-BCLX | Cell Singanlling | 2764 (1:1000 for WB) |
| Rabbit-anti-BIM | Enzo Life Sciences | ADI-AAP-330-E (1:500 for WB) |
| Rabbit-anti-P21 | Abcam | 7960 (1:1000 for WB) |
| Rat-anti-BMF | Gift from Andreas Strasser | clone 17A9 (1:500 for WB) |
| Rabbit-anti-GAPDH | Cell Signalling | 2118. clone 14C10 (1:5000 for WB) |
| Mouse-anti-HSP90 | Santa Cruz | sc-13119. clone F8 (1:2000 for WB) |
| phH3 | Cell Singanlling | 9701 (1:400 for IHC) |
| HA | Cell Singanlling | 3724 (1:4000 for IHC) |

| Reagent/resource | Reference or source | Identifier or catalog number |
| --- | --- | --- |
| Cleaved Caspase 3 | Cell Singanlling | 9664 (1:1000 for IHC) |
| P21 | Abcam | 7960 (1:1000 for IHC) |
| phospho-Histone3-PE | Biolegend | Cat# 650807 clone 11D8 (1:250 for FACS)* |
| Annexin V-FITC | Biolegend | Cat# 640945 (1:1000 for FACS) |
| Annexin V-Alexa647 | Biolegend | Cat# 640912 (1:1000 for FACS)* |
| CD19-eFluor605 | Biolegend | Cat# 115540 clone 6D5 (1:400 for FACS)* |
| CD45R/B220-eFluor510 | Biolegend | Cat# 103247 RA3-6B2 (1:400 for FACS)* |
| CD45R/B220-eFluor421 | Biolegend | Cat# 562922 RA3-6B2 (1:400 for FACS)* |
| IgM-eFluor450 | eBioscience | Cat# 48-5890-82 clone eB121-15F9 (1:400 for FACS)* |
| IgM-APC/Cy7 | Biolegend | Cat# 406516 RMM-1 (1:400 for FACS)* |
| IgM-PE/Cy7 | Biolegend | Cat# 406514 clone RMM-1 (1:400 for FACS)* |
| IgD-PerCP/Cy5.5 | Biolegend | Cat# 405710 clone 11-26 C.2 A (1:400 for FACS)* |
| IgA-PE | eBioscience | Cat# 12-4204-81 clone mA-6E1 (1:400 for FACS)* |
| CD93/AA4.1-PE/Cy7 | Biolegend | Cat# 136506 clone AA4.1 (1:300 for FACS) *BM |
| CD3-PE/Cy7 | eBioscience | Cat# 25-0031-82 clone 145-2C11 (1:400 for FACS)* |
| CD4-APC/Cy7 | Biolegend | Cat# 100414 clone GK1.5 (1:400 for FACS)* |
| CD4-PerCP/Cy5.5 | eBioscience | Cat# 45-0042-82 clone RM4-05(1:400 for FACS)* |
| CD4-BV421 | eBioscience | Cat# 48-0042-82 clon RM4-5 (1:400 for FACS)* |
| CD8-PE | Biolegend | Cat# 100708 clone 53-6.7 (1:400 for FACS)* |
| CD8-PE/Cy7 | Biolegend | Cat# 100722 clone 53-6.7 (1:400 for FACS)* |
| CD8-Alexa647 | BD | BD 557682 clone 53-6.7 (1:400 for FACS) |
| CD44-eFluor510 | Biolegend | Cat# 103044 clone IM7 (1:400 for FACS)* |
| CD25-PE | Biolegend | Cat# 102008 clone PC61 (1:300 for FACS)* |
| CD25-eFluor421 | Biolegend | Cat# 102043 clone PC61 (1:300 for FACS)* |
| CD117/ckit-PE/Cy7 | Biolegend | Cat# 105814 clone 2B8 (1:200 for FACS)* |
| CD117/ckit-APC | Biolegend | Cat# 105812 clone 2B8 (1:200 for FACS) |
| Gr1/Ly-6G/Ly6C-eFluor605 | Biolegend | Cat# 108439 clone RB6-8C5 (1:400 for FACS)* |
| Gr1/Ly-6G/Ly6C-PerCP/Cy5.5 | Biolegend | Cat# 108428 clone RB6-8C5 (1:400 for FACS)* |
| CD11b/Mac1-PE | Biolegend | Cat# 101208 clone M1-70 (1:400 for FACS)* |
| CD11b/Mac1APC/Cy7 | Biolegend | Cat# 105812 clone M1/70 (1:400 for FACS)* |
| CD71eFluor450 | Biolegend | Cat# 113813 cloneRI7217 (1:400 for FACS)* |

| Reagent/resource | Reference or source | Identifier or catalog number |
|---|---|---|
| Ter-119-PerCP/Cy5.5 | Biolegend | Cat# 116228 clone Ter-119 (1:300 for FACS)* |
| Sca1/Ly6A/E-PE | Biolegend | Cat# 108107 clone D7 (1:200 for FACS)* |
| CD150-FITC | Biolegend | Cat# 115916 clone TC15-12F12.2 (1:200 for FACS)* |
| CD48-PerCP/Cy5.5 | Biolegend | Cat# 103421 clone HM48-1 (1:200 for FACS)* |
| CD34-eFluor421 | Biolegend | Cat# 152208 clone SA376A4 (1:200 for FACS) |
| CD34-FITC | Milteny | Cat# 130-105-831 clone REA383 (1:200 for FACS)* |
| CD135/FLT3-Alexa647 | Biolegend | Cat# 135309 clone A2F10 (1:200 for FACS)* |
| CD62L-PE | Biolegend | Cat# 161204 clone W18021D (1:200 for FACS)* |
| CD16/32 / Fcγ R-PerCP/Cy5.5 | Biolegend | Cat# 101323 clone 93 (1:200 for FACS) |
| CD127/IL7R-PE/Cy7 | Biolegend | Cat# 135013 clone A7R34 (1:200 for FACS) |
| CD45R/B220-Bio | eBioscience | Cat# 13-0452-75 clone RA3-6B2 (1:500 for FACS)* |
| CD38-APC | Biolegend | Cat# 102712 clone 90 (1:200 for FACS)* |
| CD95/Fas-PE/Cy7 | BD | BD 557653 clone JO2 (1:300 for FACS)* |
| Ter-119-Bio | eBioscience | Cat# 13-5921-75 clone Ter-119(1:300 for FACS)* |
| CD3-Bio | eBioscience | Cat# 13-0031-75 145-2C11 (1:500 for FACS)* |
| CD11b-Bio | eBioscience | Cat# 13-0112-75 clone M1/70 (1:400 for FACS)* |
| Gr1/Ly-6G/Ly6C-Bio | eBioscience | Cat# 13-5931-75 clone RB6-8C5 (1:300 for FACS)* |
| NK1.1-Bio | Biolegend | Cat# 108704 clone PK136 (1:200 for FACS)* |
| CD45.1/Ly5.1-eFluor506 | eBioscience | Cat# 69-0453-82 (1:500 for FACS)* |
| Streptavidin-APC/Cy7 | Biolegend | Cat# 405208 (1:300 for FACS)* |
| Streptavidin-eFluor605 | Biolegend | Cat# 405229 (1:300 for FACS) |

*FACS antibodies used to monitor animals 12 months after bone marrow reconstitution shown in Fig. EV3.*

**Oligonucleotides and sequence-based reagents**

| | | |
|---|---|---|
| PCR Primer: Mycoplasma | Forward: GGG AGC AAA CAG GAT TAG ATA CCC T | Reverse: TGC ACC ATC TGT CAC TCT GTT AAC CTC |

**Chemicals. enzymes and other reagents**

| | | |
|---|---|---|
| RPMI-1640 | Sigma-Aldrich | Cat# R0883 |
| Optimem | Gibco | Cat# 31985070 |
| FCS | Sigma | F0804 (10%) |
| L-glutamine | Sigma | G7513 (2 mM) |
| Penicillin | Sigma | P0781 (100 U/ml) |
| Streptomycin | Sigma | P0781 (100 µg/ml) |
| 2-mercaptoethanol | Sigma | M3148 (50 µM) |
| IL6 | PeproTech | Lot #090850 A3013 (20 ng/ml) |
| IL3 | PeproTech | Lot #120948 C2013. (10 ng/ml) |

| Reagent/resource | Reference or source | Identifier or catalog number |
|---|---|---|
| Metafectene | biontex. | T020 |
| FLT3 | Supernatant of B16 melanoma cells (4%) | |
| CHO/SCF | Supernatant of SCF-producing WEHI-231 cells (2%) | |
| b-Estradiol | Sigma | E2758 (1 µM) |
| SPY650-DNA | Spirochrome | 1:1000 for live-cell imaging |
| MethoCult™ | Stem Cell Technologies | SF H4436 medium |
| MethoCult™ | Stem Cell Technologies | SF M3630 medium |
| TO-PRO-3 Iodide | Thermo Fisher | T3605 |
| M-CSF | PeproTech | 315-02 (50 ng/ml) |
| G-CSF | PeproTech | 250-05 (50 ng/ml) |
| Doxycycline food | SNIFF | Cat# 1861294 |
| Doxycycline | Sigma | D-9891 (1 µg/ml) |
| Neomycin | Sigma | N1876. Lot SLBT0907 (3.184 mg/ml) |
| DAPI | Sigma | D-9542 (1 µg/ml) |
| Propidium Iodide | Thermo Fisher Scientific | Cat# P1304MP (40 ng/ml for cell cycle analysis; 1 µg/ml for live-cell imaging) |
| BV staining buffer | BD Horizon | Cat# 566349 (1:5) |
| Albumin Bovine Serum. Fractin V | Calbiochem | Cat# 12659 |
| Triton X-100 | Merck | Cat# 108603 (0.25% for intracellular phH3 staining) |
| Red blood cell lysis buffer | 155 mM NH4Cl. 10 mM KHCO3. 0.1 mM EDTA; pH 7.5 | |
| Mayers Hematoxylin | Merck | Cat# 109249 |

**Software**

| | | |
|---|---|---|
| ImageJ/Fiji | https://imagej.net/ij/index.html | |
| GraphPad Prism9 | https://www.graphpad.com/ | |
| AneuFinder (v1.10.1) | Bakker et al, 2016 | |
| Bowtie2 (v2.2.4) | Langmead and Salzberg, 2012 | |
| NextSeq 500 | Illumina, San Diego, CA | |
| FlowJo® v10 | https://www.flowjo.com | |

**Other**

| | | |
|---|---|---|
| 70-µm strainer | Corning, Cambridge, MA, USA | Cat# 352350 |
| LSR Fortessa | BD | |
| ABC-Kit | Vector Laboratories | Cat# PK-6100 |
| ImmPACT kit | Vector Laboratories | Cat# SK-4105 |
| Microtome | Leica Jung RM 2035 | |
| S3 microscope | Sartorius, Göttingen, Germany | |
| Illumina NextSeq 500 | Illumina, San Diego, CA | |

## Animals used and generation of bone marrow chimeras

Animal experiments were performed in accordance with Austrian legislation (BMWF: 66-011/0106-WF/3b/2015 and GZ: 2021-0.222.1888). The generation of *Vav-BCL2* transgenic (Ogilvy et al, 1999), *Bcl2l11/Bim*[−/−] (Bouillet et al, 1999), *Bbc3/Puma*[-/-] (Villunger et al, 2003), *Pmaip/Noxa*[−/−] (Villunger et al, 2003), GFP-reporter (Premsrirut et al, 2011) and *Mad2*-transgenic animals (Rowald et al, 2016) were described before. All mice were maintained or backcrossed on a C57BL/6N genetic background or backcrossed for at least 6 generations in case of *Mad2*-transgenic mice. Mice were housed in ventilated cages with nesting material and were maintained on a 12:12-h light:dark cycle.

Bone marrow reconstitutions were performed using C57BL6/N Ly5.1 recipient mice at the age of 8–10 weeks. Mice were irradiated with a single dose of 10 Gy and reconstituted with 200 μl PBS containing $4 \times 10^6$ total bone marrow cells of Ly5.2 or Ly5.1/2 donor origin, injected via the tail vein. Mice were given Neomycin for 2–3 weeks in the drinking water before swapping to regular water. Mice were set directly (50:50 chimeras) or 12 weeks after reconstitution on Dox food. Chimaeras were analyzed after an additional 10 weeks of transgene induction. Whole body MAD2 overexpressing mice were analyzed at age 15–20 weeks indiscriminate of sex. Animals were examined once terminally sick (losing >17% of initial body weight) or after surviving for 17 days on Dox food. The genotype was blinded for the experimenter. Peripheral blood was sampled from the facial vein and analyzed using a ScilVet Abc blood counter.

## Generation of HoxB8 progenitor cells

Cell lines were generated as described in (Schuler et al, 2019); in short, fresh bone marrow was cultivated in Optimem supplemented with recombinant mouse IL3 and IL6, spin-infected with *HoxB8*-encoding retrovirus and cultivated in RPMI-1640 complete (PFs) or Optimem complete (PNs) with either FLT3 (PF) or SCF (PN) supernatant. Primary cell culture was regularly checked for mycoplasma contamination with the following primers: Forward: GGG AGC AAA CAG GAT TAG ATA CCC T; Reverse: TGC ACC ATC TGT CAC TCT GTT AAC CTC.

## Immunoblotting and Immunoprecipitation

Cells lysis, immunoblot analyses, and immunoprecipitation were performed as previously described (Haschka et al, 2020); used antibodies are listed in the Reagents and Tools table.

## Flow cytometry and intracellular staining for phH3

Single-cell suspensions of the spleen, thymus and lymph node were achieved by smashing the whole organ through a 70-μm strainer in PBS/2% FCS. Single-cell suspensions of the spleen were pelleted and resuspended in 1 ml red blood cell lysis buffer for erythrocyte depletion for 2 min at RT, washed with PBS, then stained with antibodies and quantified for cellularity using a hemocytometer (Neubauer) and trypan blue exclusion. Flow cytometric analysis of single-cell suspensions was performed on an LSR Fortessa (BD) and analyzed using FlowJo® v10 software.

HoxB8 cells were fixed in 70% ethanol and stored at −20 °C. After two washes with PBS, cells were incubated for 15 min in PBS/0.25% Triton X-100 on ice for permeabilization. Cells were washed in PBS/1% BSA and incubated for 60′ with anti-mouse phospho-Histone3 S10 mAb. Cells were washed and stained with ToPro-3 (100 nM) for DNA content analysis. HoxB8 cell were stained with Annexin V in Annexin-V binding buffer and DAPI. Subsequently, the percentage of apoptotic/dead cells was analyzed by flow cytometry. List of Antibodies can be found in the Reagents and Tools table.

## Single-cell whole-genome sequencing (scWGS) and data analysis

HoxB8-PN cell lines treated with Doxycycline were harvested at different time points and frozen in freezing media (FCS + 10% DMSO) until further processing for whole-genome single-cell sequencing, performed on 24 cells/cell line using a NextSeq 500 (Illumina, San Diego, CA). Data analysis was performed as described previously (Sladky et al, 2020). Short, the sequencing data was aligned using the murine reference genome (GRCm38) along with the Bowtie2 (v2.2.4) software (Langmead and Salzberg, 2012). The AneuFinder (v1.10.1) (Bakker et al, 2016) was used for copy number variation analysis; libraries were processed, including GC correction, elimination of regions prone to artefacts, and validation of mappability. The divisive copy number calling algorithm was employed with a bin size of 1 Mb. The modal copy number state was determined based on the anticipated ploidy state. To calculate the aneuploidy score for each library, the average absolute deviation from the expected euploid copy number was computed per bin. The corresponding data has been submitted to the European Nucleotide Archive (ENA) and can be accessed using this number (PRJEB72260) https://www.ebi.ac.uk/ena/browser/home.

## Live-cell imaging

HoxB8 cells were seeded in media with or without 1 μg/ml Doxycycline and 1 μg/ml PI and imaged every 2 h in an Incucyte S3 microscope using the ×10 objective (Sartorius, Göttingen, Germany). The analysis function of the system was used to calculate the red object count (ROC), representing PI[+] cells per mm², normalized to the area covered by the cells at start of imaging. Number of positive cells at the imaging start ($t = 0$ h) was subtracted from each time point.

For experiments shown in Fig. 1G,H, HoxB8-PF cells were seeded in media containing 1% methylcellulose in a 35-mm Petri dish at a density of 400.000/ml and treated with or without 1 μg/ml Doxycycline and SPY650-DNA stain. Image acquisition was done with Zeiss Axio Observer microscope with ×63 objective every 10 min in an environmental chamber set to 37 °C. A total of 40–52 cells were randomly selected, and mitotic duration and cell fate were assessed manually using Fiji. The clear appearance of metaphase defined the onset of mitosis.

## Colony-forming assay

For colony-forming assays, MethoCult™ SF H4436 medium was used. Bone marrow was seeded with or without 1 μg/ml Doxycycline at a

density of $10^3$ cells/ml/35-mm cell culture dish and incubated in the presence of serum-free medium with mouse cytokines: M-CSF for macrophages and G-CSF for granulocytes. MethoCult™ SF M3630 medium with IL-7 was used for B-cell outgrowth. Colonies were counted 10–14 days post-seeding.

## Histology and immunohistochemistry

The intestine was transferred as a "Swiss roll" to PBS/4% paraformaldehyde (PFA) for fixation and embedding. After 24 h of fixation, tissue was transferred to 70% EtOH. Dehydration and processing were performed overnight via Shandon Citadel 1000 (50% EtOH, 70% EtOH, 80% EtOH, 2 × 96% EtOH, 2 × 100% EtOH, 2× Xylene), and then the tissue was embedded in paraffin. Histology blocks were cut sagittal with a rotation microtome, and sections were placed on silanized glass slides. Slides were dried for 30–60 min at 60 °C. After deparaffinisation and rehydration of the slides, antigen retrieval was performed in 10 mM NaCit-buffer, pH 6 at 90 °C. After washing with TBST, endogenous peroxidase activity was blocked with 3% $H_2O_2$ followed by permeabilization and blocking of non-specific binding (5% goat serum in TBST) for 1 h. In total, 50 µl of the primary antibody (see Reagents and Tools table) was added to the slides and incubated at 4 °C overnight. Slides were washed with TBST, before incubation with the biotinylated secondary antibody (goat anti-rabbit 1:100; Dianova 111-065-144), for 2 h at RT. Amplification was performed with the Vectastain ABC-Kit for 30 min at RT followed by washing with TBST. Peroxidase substrate mix was prepared according to the ImmPACT kit manual, 100 µl of the solution was added onto each slide. Counterstaining was done with Mayer hematoxylin (). After that, slides were dehydrated and transferred to Xylene, then mounted with Histokitt. Images were taken with Zeiss Axioplan2 imaging at ×10 and ×20 magnification.

## Histopathological examination

In all, 4-µm sections of paraffin-embedded H&E-stained intestine were examined for histological analysis by a board-certified pathologist. Grading was done according to the following definitions: Grade 1 = increased apoptotic enterocytes and neutrophilic granulocytes in the *L. epithelialis*; Grade 2 = Grade 1 plus: crypt abscesses (colon); increased inflammatory infiltrate of the *L. propria* (ileum); Grade 3 = Grade 1 plus 2: erosions of the *L. epithelialis*; Grade 4: Grade 1–3 plus: ulceration.

## Immunohistochemistry scoring

The number of cells positive for phH3 or active Caspase 3 was assessed manually, with ImageJ, in at least 5–10 (colon) or 15–20 (small intestine) randomly chosen fields at ×20 magnification per mouse. Images were acquired on a Zeiss Axioplan2 imaging microscope using ZENblue software. A mean is calculated for each mouse and genotype. All images were blinded before analysis.

## Statistical analysis

All statistical analyses were performed using Prism9 software (GraphPad, La Jolla, CA, USA). One-tailed paired Student's *t* test or analysis of variance (ANOVA) for multiple group comparisons. Data are shown as Mean ± SEM if not stated otherwise. Statistical significance is shown with symbols: *$P$ value < 0.05, **$P$ value < 0.01, ***$P$ value 0.001, ****$P$ value < 0.0001. Only conclusion-relevant significant differences, or lack thereof, are marked.

## Data availability

Single-cell whole-genome sequencing (scWGS) data has been submitted to the European Nucleotide Archive (ENA) with the accession number PRJEB72260. Source data for Figs. 1–6 can be found online on BioStudies, accession number. S-BSST1371: https://www.ebi.ac.uk/biostudies/studies/S-BSST1371?key=8d70648b-3274-4909-8566-74f2c9cf9831.

The source data of this paper are collected in the following database record: biostudies:S-SCDT-10_1038-S44319-024-00160-3.

## Peer review information

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

## Acknowledgements

The authors are grateful to I Gaggl, C Soratroi, J Heppke, M Fischer, N Heinrich, and M Saurwein for excellent technical assistance or animal care. The authors also want to thank A Strasser and L Fava for sharing mice or reagents and all lab members for fruitful discussion. This research was funded in part by the Austrian Science Fund (FWF) [10.55776/I6642], part of the DFG-funded TRR353 initiative, and FWF grants [10.55776/I3271] & [10.55776/P36658], as well as the European Research Council, ERC, AdG POLICE (787171). F Eichin acknowledges support from the DOC fellowship program of the Austrian Academy of Sciences (ÖAW). For open access purposes, the corresponding author has applied a CC BY public copyright license. We acknowledge BioRender to be used in some figures, including the synopsis thumbnail.

## Author contributions

**Gerlinde Karbon**: Data curation; Formal analysis; Investigation; Visualization; Methodology; Writing—original draft; Writing—review and editing. **Fabian Schuler**: Data curation; Formal analysis. **Vincent Z Braun**: Formal analysis; Investigation. **Felix Eichin**: Formal analysis; Methodology. **Manuel Haschka**: Investigation; Methodology. **Mathias Drach**: Formal analysis; Investigation. **Rocio Sotillo**: Resources; Methodology; Writing—review and editing. **Stephan Geley**: Data curation; Methodology; Writing—review and editing. **Diana CJ Spierings**: Data curation; Formal analysis; Methodology. **Andrea E Tijhuis**: Data curation; Formal analysis; Methodology. **Floris Foijer**: Data curation; Methodology; Writing—review and editing. **Andreas Villunger**: Conceptualization; Data curation; Supervision; Funding acquisition; Writing—original draft; Project administration; Writing—review and editing.

Source data underlying figure panels in this paper may have individual authorship assigned. Where available, figure panel/source data authorship is listed in the following database record: biostudies:S-SCDT-10_1038-S44319-024-00160-3.

## Disclosure and competing interests statement

The authors declare no competing interests.

# Expanded View Figures

**Figure EV1.   Characterization of cellular responses to MAD2 overexpression.**

HoxB8 cell lines were generated from the bone marrow of mice of the indicated genotype treated with ($+$) or without (-) Doxycycline for the indicated times. (**A**) HoxB8-PF cells were harvested after 48 h of transgene induction with different Doxycycline concentrations for immunoblot analysis using the indicated antibodies. (**B**) HoxB8-PF cells were treated with 1 µg/ml Doxycycline or left untreated for the indicated times and harvested for immunoblot analysis using the indicated antibodies. (**C, D**) Gating strategy used in flow cytometric analyses of cells shown in Fig. 1C,D. (**E**) HoxB8-PN cells were incubated with and without Doxycycline (1 µg/ml). Cell death was followed in Incucyte live-cell microscopy by PI uptake; pictures were taken every 2 h. MR ($n = 3$; 2 biological replicates), MR2 ($n = 5$; 3 biological and 1–2 technical replicates); data shown as mean ± SEM. (**F, G**) (**F**) HoxB8-PN cells were analyzed for GFP reporter expression before and during Doxycycline (1 µg/ml) treatment in PI-negative cells by FACS (MRG $n = 4$, MRG2 $n = 1$) and on (**G**) protein level (*: indicates residual HA-MAD2 signal after re-probing the membrane with anti-GFP antibody). (**H**) HoxB8-PF cells were analyzed for dead cells (Annexin V/DAPI) and GFP reporter expression before and during Doxycycline (1 µg/ml) treatment by FACS. MRG ($n = 4$, 2 biological and 2 technical replicates), RG ($n = 6$, 3 biological and 2 technical replicates); data shown as mean ± SEM. (**I**) HoxB8-PFs were sorted for the expression of GFP (negative, low and high) after 24 h of Doxycycline (1 µg/ml) treatment and transgene induction and subjected for immunoblot analysis using indicated antibodies. Data information: (**E, F, H**) Data shown as mean ± SEM. (**H**) Two-way ANOVA, Sidak's multiple comparisons. ns not significant, *$P \leq 0.05$,

                                                

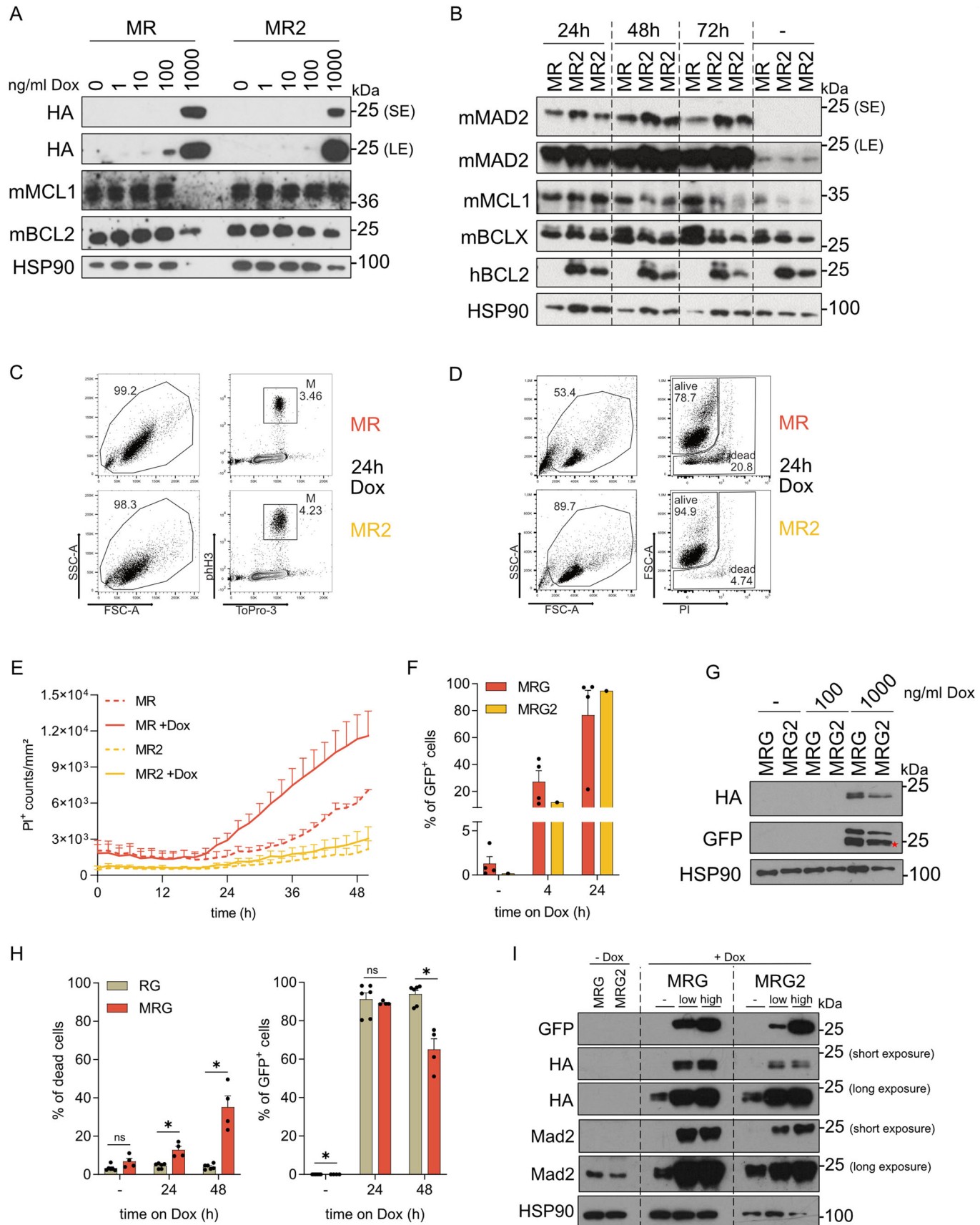

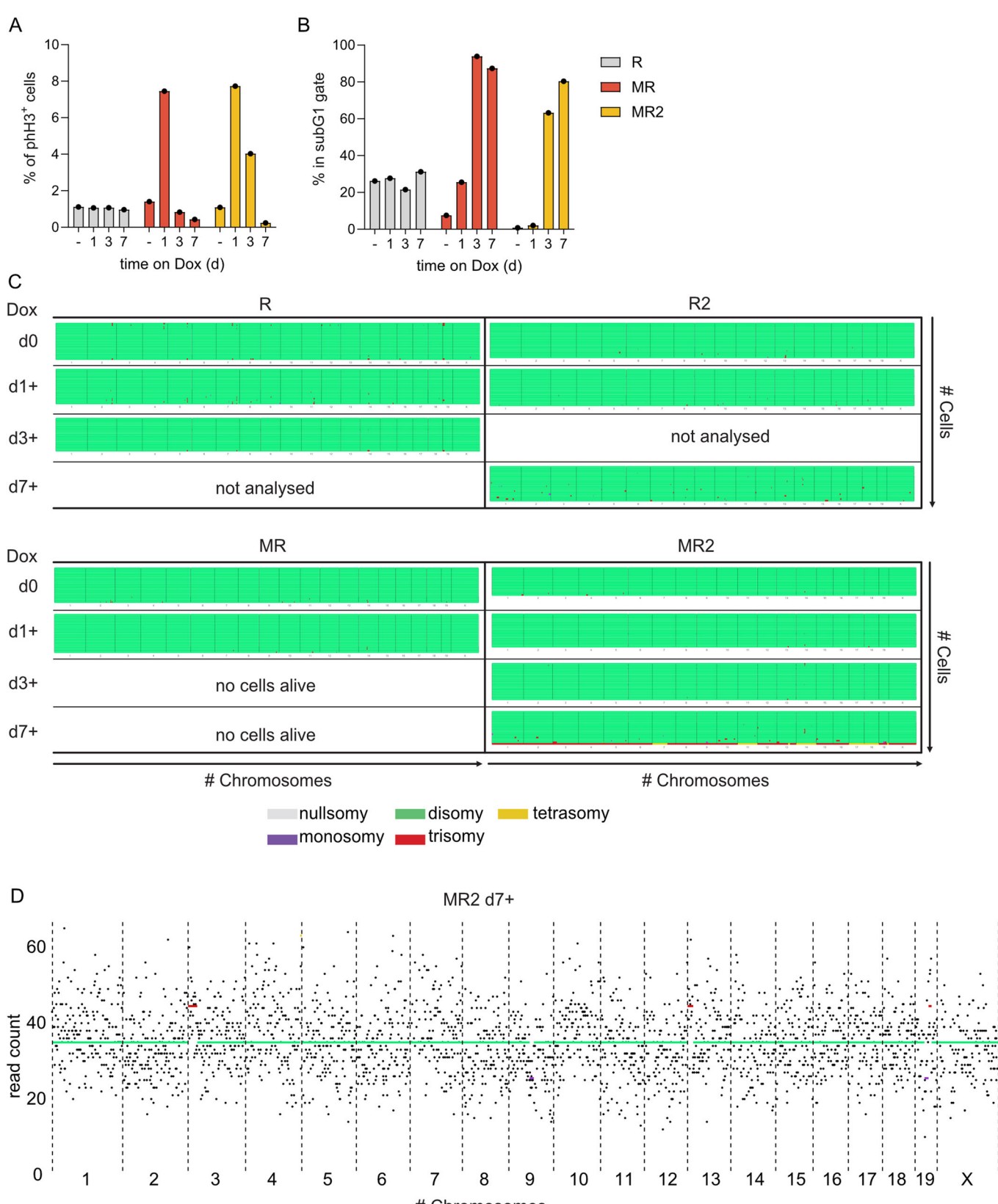

**Figure EV2. HoxB8 cells do not show CIN upon MAD2 overexpression.**

(A, B) Cells used for scWGS were fixed and analyzed for quantification of (A) mitotic cells (phH3⁺) and (B) viability (sub-G1 staining) by FACS (n = 1). (C) scWGS analyses of cells in (A, B) over time. (D) Representative read count of MR2 cell line 7 days on Doxycycline with euploid karyotype.

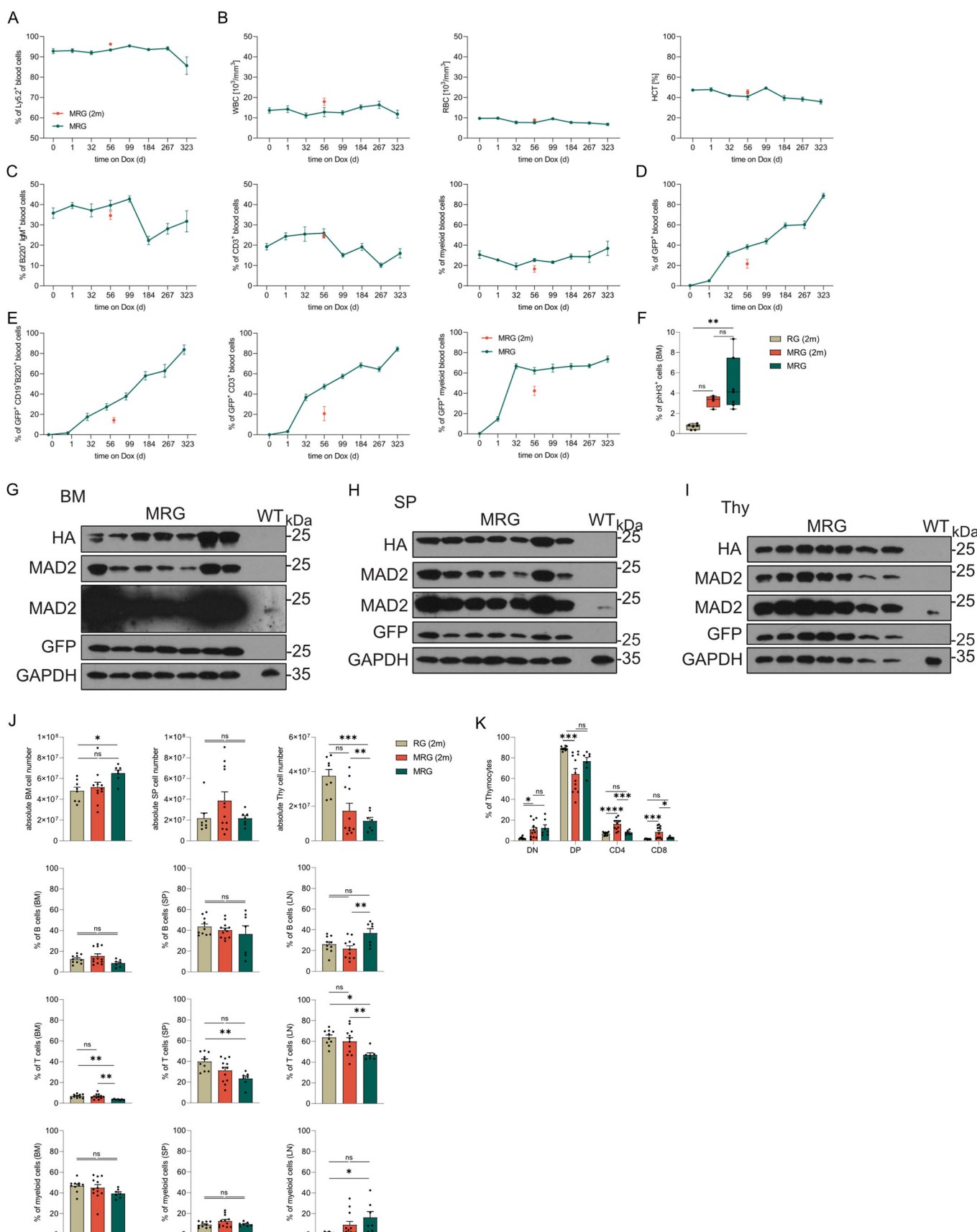

◄

**Figure EV3.   Long-term MAD2 overexpression does not trigger blood cancer.**

Ly5.1$^+$ animals were reconstituted with BM derived from Ly5.2$^+$ MRG donors, set on Dox food after 12 weeks of reconstitution and monitored over 12 month (MRG n = 7). Data of reconstituted animals from our first cohort, analyzed after 2 months (2 m) on Dox food (shown in Fig. 4 and Appendix Figure S3), is displayed again for reference purposes here, RG (2 m) $n$ = 6–10 and MRG (2 m) $n$ = 4–12. (**A**) Peripheral blood was stained to quantify the percentage of Ly5.2$^+$ cells. (**B**) White (WBC), red blood cell (RBC) counts and hematocrit (HCT) were determined using a SciVet abc blood cell analyser. (**C**) Analysis of mature B cells (B220$^+$IgM$^+$), T cells (CD3$^+$) and myeloid cells (Gr1$^{+/-}$CD11b$^+$) in peripheral blood over time. (**D, E**) GFP expression in total blood and individual blood cell types. (**F**) Bone marrow (BM) was stained using antibodies specific for phH3 to identify mitotic cells. (**G–I**) (**G**) BM, (**H**) spleen (SP) and (**I**) thymus (Thy) of reconstituted mice were analyzed for MAD2 transgene expression after 12 month on Dox-containing food by western blot (WT: non-reconstituted WT mouse). (**J**) Cellularity of BM, SP and Thy as well as leukocyte content were analyzed and compared to that seen in mice 2 month after reconstitution. Cell surface marker-specific antibodies were used to identify total B cells (CD19$^+$B220$^+$), T (CD4$^+$&CD8$^+$) and myeloid cells (Gr1$^{+/-}$CD11b$^+$). (**K**) Thymocytes were stained with antibodies specific for CD4 and CD8 to identify DN (CD4$^-$CD8$^-$), DP (CD4$^+$CD8$^+$), T helper (CD4$^+$CD8$^-$) and T cytotoxic (CD4$^-$CD8$^+$) thymocyte subsets. Data information: (**A–E, J, K**) Data shown as mean ± SEM. (**F**) Data shown as Min to Max with median and IQR of: RG (2 m) 0.5, MRG (2 m) 0.78, MRG 4.64. (**F, J**) One-way ANOVA, Tukey's multiple comparisons. (**K**) One-way ANOVA within each subset (DN, DP, CD4, CD8). ns not significant, *$P$ ≤ 0.05, **$P$ ≤ 0.01, ***$P$ ≤ 0.001, ****$P$ ≤ 0.0001.

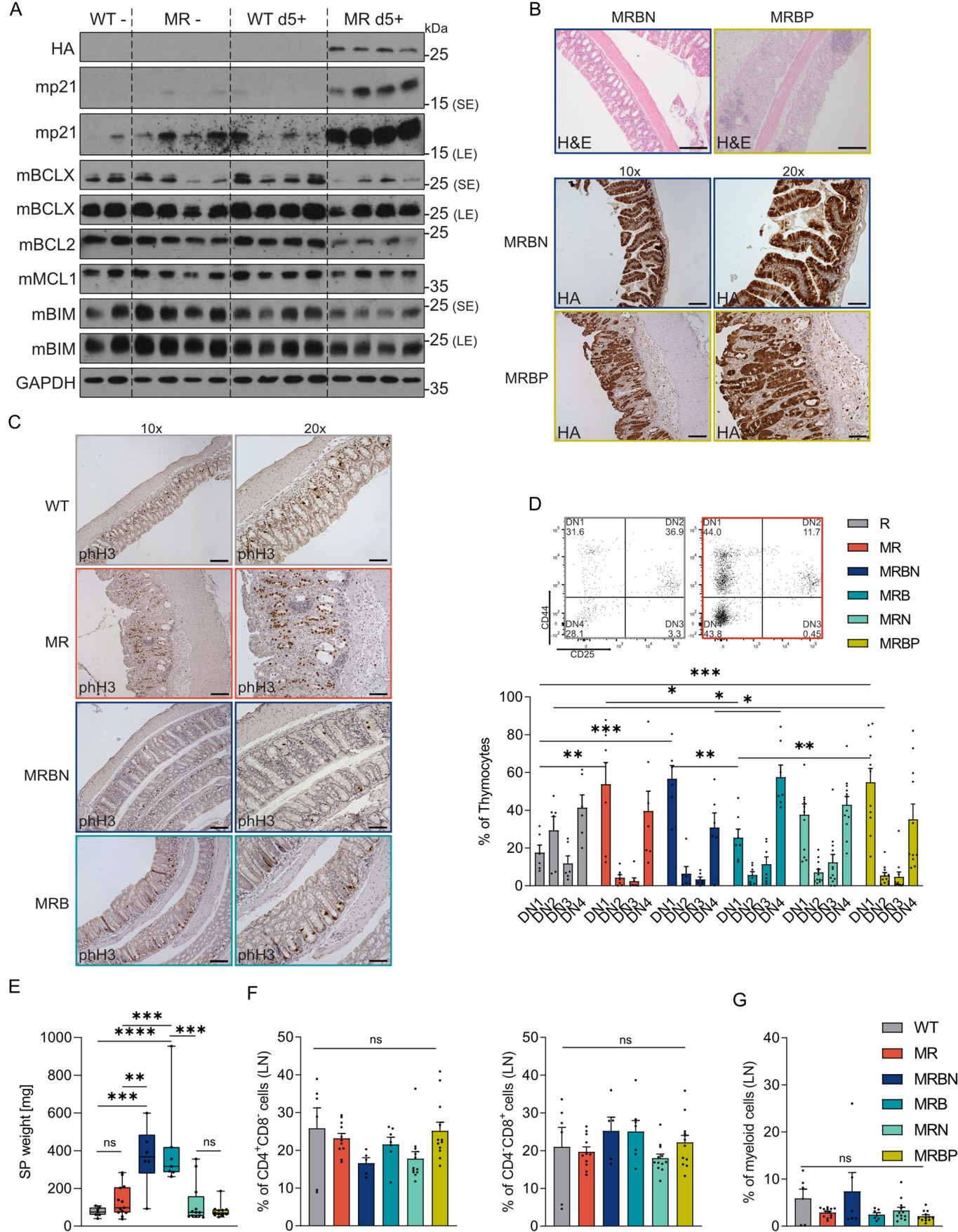

**Figure EV4. MAD2 overexpression causes gastrointestinal syndrome.**

(A) Colonic extracts of mice of the indicated genotypes were subjected to immunoblotting using the indicated antibodies. (B) H&E and HA staining of paraffin-embedded sections from Swiss rolls of the small intestine (SI) of MRBN and MRBP animals placed on Dox food; scale bar for H&E staining 200 μM, scale bar for HA staining 1 mm. (C) Swiss roll sections of the colon of mice with indicated genotypes were analyzed for the presence of mitotic cells (phH3$^+$); scale bar 1 mm. (D) Gating strategy used in flow cytometric analyses of CD4$^-$CD8$^-$double-negative (DN) thymocytes stained with antibodies for cell surface markers discriminating stages DN1 (CD44$^+$CD24$^-$), DN2 (CD44$^+$CD24$^+$), DN3 (CD44$^-$CD24$^+$) and DN4 (CD44$^-$CD24$^-$). (E) Spleen (SP) weight from healthy and terminally sick animals. (F, G) Lymph nodes (LN) stained for helper (CD4$^+$) or cytotoxic (CD8$^+$) T cells and myeloid cells (Gr1$^{+/-}$CD11b$^+$). WT ($n = 6$–7), MR ($n = 7$–13), MRBN ($n = 5$–6), MRB ($n = 7$), MRN ($n = 12$) and MRBP ($n = 11$–12). Data information: (D, F, G) Data shown as mean ± SEM. (E) Data shown as Min to Max with median and IQR of: WT 76.5, MR 152, MRBN 386, MRB 252.5, MRN 68, MRBP 29. (D) Two-way ANOVA, Tukey's multiple comparisons. (E–G) One-way ANOVA, Tukey's multiple comparisons. *$P \leq 0.05$, **$P \leq 0.01$, ***$P \leq 0.001$, ****$P \leq 0.0001$.

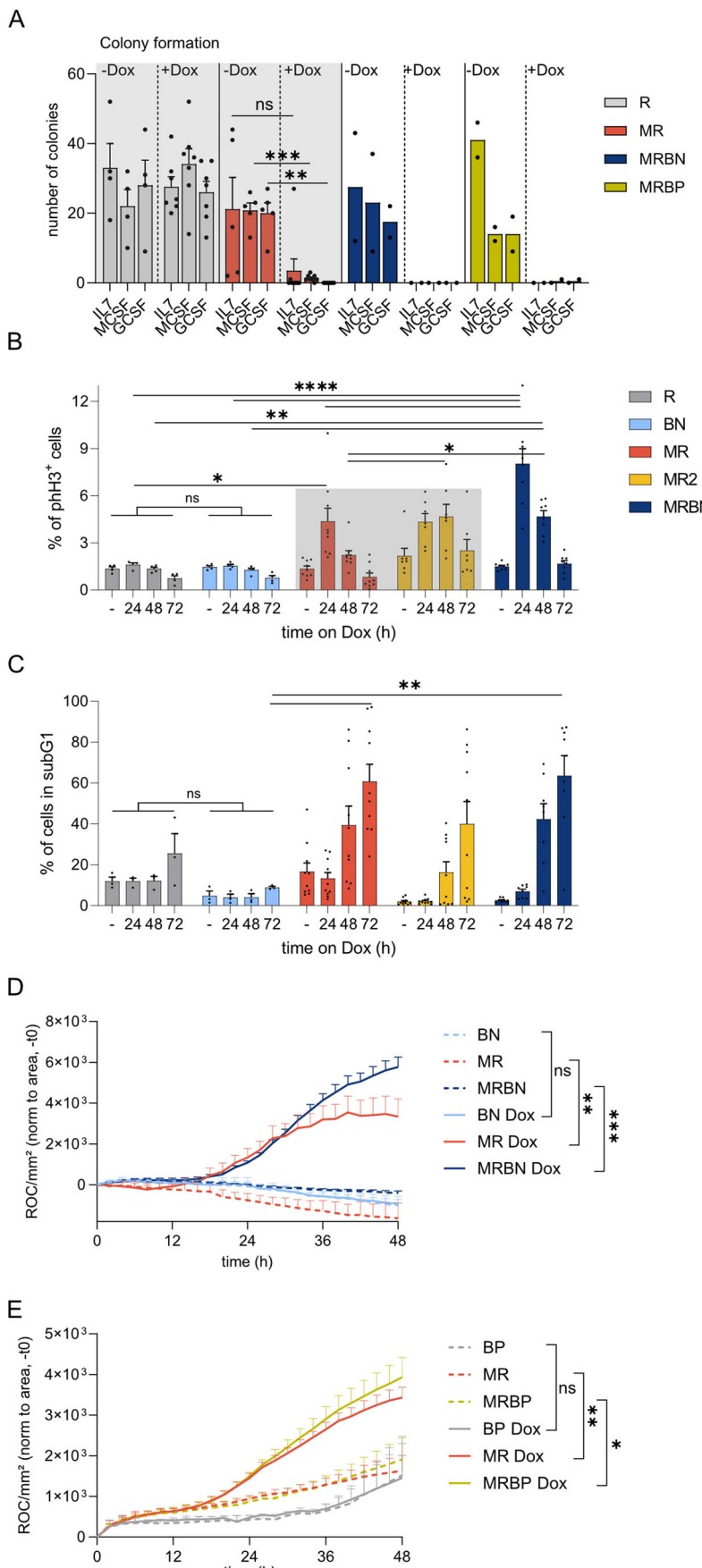

**Figure EV5. BH3-only protein loss fails to rescue blood cells from mitotic cell death.**

(A) Colony formation potential of R, MR, MRBN and MRBP transgenic bone marrow was assessed from individual mice in methylcellulose assays in the presence (+) or absence (-) of Doxycycline. (Data points of R and MR, ± Dox of Fig. 3A are shown again for reference). R+Dox ($n = 7$), MR+Dox ($n = 8$), R-Dox ($n = 4$), MR-Dox ($n = 5$). MRBN ($n = 2$), MRBP ($n = 2$). (B, C) HoxB8-PF cells established from Mad2 transgenic animals were fixed and analyzed by flow cytometry for the presence of mitotic (B, phH3⁺) and dead (C, sub-G1) cells (Data points of MR and MR2 cells of Fig. 1C are shown again for reference). R ($n = 2$ biological replicates, 1 measured three times), MR ($n = 9$; 6 biological and 1–2 technical replicates), MR2 ($n = 7$; 3 biological and 1–3 technical replicates), BN ($n = 4$ biological replicates), MRBN ($n = 8$, 6 biological and 1–3 technical replicates); mean ± SEM. (D) HoxB8-PF cells were incubated with and without Doxycycline (1 µg/ml). Cell death was recorded over time by Incucyte live-cell microscopy and PI uptake; pictures were taken every 2 h. BN ($n = 5$; 1 biological and 5 technical replicates), MR ($n = 5$; 1 biological and 5 technical replicates), MRBN ($n = 5$; 1 biological and 5 technical replicates). (E) HoxB8-PN cells were incubated with and without Doxycycline (1 µg/ml). Cell death was recorded over time by Incucyte live-cell microscopy and PI uptake; pictures were taken every 2 h. BP ($n = 2$, 1 biological and 2 technical replicates), MR ($n = 6$, 4 biological and 2 technical replicates), MRBP ($n = 4$, 2 biological and 2 technical replicates). Data information: (A–E) Data shown as mean ± SEM. (A) Unpaired $t$ test, Welchs correction. _*(D, E) Unpaired $t$ test, Welchs correction for 48 h time point. *_ (B, C) Two-way ANOVA, Tukey's multiple comparisons. *$P \le 0.05$, **$P \le 0.01$, ***$P \le 0.001$, ****$P \le 0.0001$.

