## [Peer Review File · EMBO Reports]

Chronic spindle assembly checkpoint activation causes myelosuppression and gastrointestinal atrophy

Andreas Villunger, Gerlinde Karbon, Fabian Schuler, Manuel Haschka, Mathias Drach, Rocio Sotillo, Stephan Geley, Diana Spierings, Andrea Tijhuis, Floris Fojjer, Vincent Braun, and Felix Eichin

Corresponding author(s): *Andreas Villunger (andreas.villunger@i-med.ac.at)*

Review Timeline:

Submission Date:	7th Sep 23
Editorial Decision:	20th Oct 23
Revision Received:	25th Mar 24
Editorial Decision:	19th Apr 24
Revision Received:	30th Apr 24
Accepted:	6th May 24

Editor: Achim Breiling

Transaction Report:

Dear Prof. Villunger,

Thank you for the submission of your research manuscript to EMBO reports. I have now received the reports from the three referees that were asked to evaluate your study, which can be found at the end of this email.

As you will see, the referees think that the findings are of interest. However, the referees have several comments, concerns, and suggestions, indicating that a major revision of the manuscript is necessary to allow publication of the study in EMBO reports. As the reports are below, and all the referee concerns need to be addressed, I will not detail them here.

Given the constructive referee comments, I would like to invite you to revise your manuscript with the understanding that all referee concerns must be addressed in the revised manuscript or in a detailed point-by-point response. Acceptance of your manuscript will depend on a positive outcome of a second round of review. It is EMBO reports policy to allow a single round of revision only and acceptance of the manuscript will therefore depend on the completeness of your responses included in the next, final version of the manuscript.

- 1) a .docx formatted version of the final manuscript text (including legends for main figures, EV figures and tables), but without the figures included. Figure legends should be compiled at the end of the manuscript text.
- 2) individual production quality figure files as .eps, .tif, .jpg (one file per figure), of main figures (up to 8) and EV figures. Please upload these as separate, individual files upon re-submission.

- 4) a complete author checklist, which you can download from our author guidelines (<https://www.embopress.org/page/journal/14693178/authorguide>). Please insert page numbers in the checklist to indicate where the requested information can be found in the manuscript. The completed author checklist will also be part of the RPF.

- 5) that primary datasets produced in this study (e.g. RNA-seq, ChIP-seq, structural and array data) are deposited in an

appropriate public database. If no primary datasets have been deposited, please also state this in a dedicated section (e.g. 'No primary datasets have been generated and deposited'), see below.

The accession numbers and database should be listed in a formal "Data Availability" section (placed after Materials & Methods) that follows the model below. This is now mandatory (like the COI statement). Please note that the Data Availability Section is restricted to new primary data that are part of this study. This section is mandatory. As indicated above, if no primary datasets have been deposited, please state this in this section

Data availability

8) Regarding data quantification and statistics, please make sure that the number "n" for how many independent experiments were performed, their nature (biological versus technical replicates), the bars and error bars (e.g. SEM, SD) and the test used to calculate p-values is indicated in the respective figure legends (also for potential EV figures and all those in the final Appendix). Please also check that all the p-values are explained in the legend, and that these fit to those shown in the figure. Please provide statistical testing where applicable. Please avoid the phrase 'independent experiment', but clearly state if these were biological or technical replicates. Please also indicate (e.g. with n.s.) if testing was performed, but the differences are not significant. In case n=2, please show the data as separate datapoints without error bars and statistics. See also: <http://www.embopress.org/page/journal/14693178/authorguide#statisticalanalysis>

9) Please add scale bars of similar style and thickness to all the microscopic images, using clearly visible black or white bars (depending on the background). Please place these in the lower right corner of the images themselves. Please do not write on or near the bars in the image but define the size in the respective figure legend.

10) Please also note our reference format:

12) We now use CRedit to specify the contributions of each author in the journal submission system. CRedit replaces the author contribution section. Please use the free text box to provide more detailed descriptions and do not provide your final manuscript text file with an author contributions section. See also our guide to authors: <https://www.embopress.org/page/journal/14693178/authorguide#authorshipguidelines>

13) We would encourage you to use 'Structured Methods', our new Materials and Methods format. According to this format, the

Materials and Methods section should include a Reagents and Tools Table (listing key reagents, experimental models, software and relevant equipment and including their sources and relevant identifiers) followed by a Methods and Protocols section in which we encourage the authors to describe their methods using a step-by-step protocol format with bullet points, to facilitate the adoption of the methodologies across labs. More information on how to adhere to this format as well as downloadable templates (.doc or .xls) for the Reagents and Tools Table can be found in our author guidelines (section 'Structured Methods'):

14) Please order the manuscript sections like this, using these names:

Title page - Abstract - Keywords - Introduction - Results - Discussion - Materials and Methods - Data availability section - Acknowledgements - Disclosure and Competing Interests Statement - References - Figure legends - Expanded View Figure legends

I look forward to seeing a revised version of your manuscript when it is ready. Please let me know if you have questions or comments regarding the revision.

Yours sincerely,

Referee #1:

In this work by Karbon et al the in vivo impact of long term activation of the spindle assembly checkpoint is investigated. The authors achieve long term SAC activation by DOX induced overexpression of Mad2 in mice and observe bone marrow aplasia and intestinal atrophy. Furthermore the authors investigate how pro-apoptotic proteins contribute to in vivo phenotypes and find that BIM co-depletion prevents intestinal atrophy but not myelosuppression. Overall this is a very carefully conducted study that provides important insight into in vivo consequences of prolonged SAC activation. I have some minor suggestions that would in my opinion strengthen the impact of the work.

I think it is important that the authors state how much additional Mad2 is expressed. I think this information is in Fig. 1B but should be quantified. In Fig. 1B they should also probe for Cdc20 to establish how much additional MCC is likely generated. The level of Mad2 over expression should be compared to some of the previous studies from Benezra that they compare their work to. Also how do these levels compare to the elevated levels of Mad2 reported in cancers? It could be that low levels of Mad2 expression is more detrimental than extremely high levels of over expression (which I think is what they obtain here). Is it possible for the authors to investigate cellular fate in relation to GFP expression (proxy for Mad2 levels) to address this?

Small typos:

be consistent in either cyclin or Cyclin and securin or Securin

page 3 line 67: securing should be securin

page 16 line 419: above - do they mean "consistent with the later"?

Referee #2:

Karbon et al have examined spindle assembly checkpoints in an interesting way by assessing if mitochondrial apoptosis limits the survival of cells with chronic SAC activity in order to prevent aneuploidy and CIN-related pathology. The authors further identify the pro-apoptotic protein BIM as rate limiting for apoptosis induction in the GI tract in response to SAC perturbation.

1. The authors state lines 186-187, "that the presence of BCL2 reduced cell death in mitosis (diM) from about 60% to 20% in MR2 cells", was this statistically significant and if so, this should be stated and shown in Figure 1L.
2. Relating to figure 2, no karyotypic abnormalities were detected in the presence or absence of Dox in R, R2, MR or MR2 cells nor within live MR2 cells remaining at days 3 or 7, perhaps the optimal timepoint was not assessed and 12 h or 2 days may have been optimal, since all cells were dead at day 3.
3. The bone marrow reconstitution experiments (Figure 3, 4) lack positive controls, such as reconstitution with the respective parental BM cells for comparative studies, i.e. mice which have also undergone irradiation and reconstitution. Are any of the values in Figure 3 significantly altered compared to T0 (50:50)?
4. Figure 3A the authors indicate in lines 243-244 that "pre-B and myeloid colony formation was drastically impaired upon Dox treatment", what does drastically mean, was the difference significant as no significance values are indicated in Figure 3A.

5. Figure 4A appears to be lacking all annotation.
6. Whilst dissecting the consequences of MAD2 overexpression on the haematopoietic system, the authors noted a reduction thymic cellularity. The dissection of the proportional difference in the earlier T cell progenitors was fairly rudimentary and requires further investigation, particularly since some the subpopulations are cycling while others non-cycling, i.e. staining with CD25 and CD44 to dissect out the DN 1-4 stages in T cell maturation most affected by MAD2 overexpression and thus the proliferative capacity of the progressive stages of T cell development.
7. Figure S6A-E, why is the 56 day time point data not integrated into the linear curve of each these graphs. Clearly something has happened at this time point, with a drop in all parameters analysed, yet no explanation is offered. Figure S6J, were any of the graphed values significant, if not, indicate in the text, rather than "comparable"?
8. Line 333-334, the authors indicate in relation to the MAD2 overexpression none of the animals' developed signs of malignant disease" and lines 340-341 "still MAD2 overexpression appears insufficient to promote spontaneous malignancies within 12 months" what signs were assessed, which organs, which pathology performed, there are no data to support these statements.

Referee #3:

Karbon et al. aimed to describe consequences of apoptosis after MAD2 overexpression. They describe that overexpression of MAD2 leads to a mitotic delay and cell-death, which can be reduced by expression of the pro-survival protein BCL2. Although systemic overexpression of MAD2 is not viable, authors were able to study effects of MAD2 overexpression in vivo by studying effects on hematopoietic system, by making use of a very elegant competitive reconstitution experiment. Mad2 overexpression led to myeloablation which was rescued by BCL2 overexpression, identifying a role for apoptosis in this process. Next to that, they have shown that SAC perturbation causes severe colitis and gastrointestinal syndrome which may have been a cause for the rapid death of mice with systemic overexpression of MAD2. Authors identified the pro-apoptotic BIM as an essential factor leading to decreased survival of the transgenic mice. However, deleting BIM did not lead to a rescue of the MAD2 overexpression phenotype in the hematopoietic system. At this point it is not clear why these effects differ in each organ system or which other factors play a role in this.

The manuscript describes interesting results which could contribute to understanding the effect of SAC perturbation on the hematopoietic system and GIT, however, some major concerns regarding the manuscript are listed below.

Main comments:

- 1) The authors set out to study the molecular players mediating cell death in response to chronic SAC deregulation in vivo. Although authors do study this partly, their manuscript would greatly benefit from a more elaborate study of additional apoptotic effectors to truly understand the role of apoptosis after MAD2 overexpression. Where BIM is an essential factor regulating effects on GIT, it does not have an effect on the hematopoietic system, where overexpression of BCL2 had a rescue effect. Why did the authors not study effect of BCL2 in the GIT? (In addition to that it is very unclear which factors and combinations were effectively tested, see comment below). In their discussion authors mention that differential effect of BIM and NOXA on hematopoietic system may be due to other apoptotic effectors BMF and BID, (also MCL1?), which should be tested in order to clarify this discrepancy and to answer their research question.
- 2) For their experiments authors make use of doxycycline which is a well-known anti-inflammatory agent. Authors fail to accurately describe effects of doxycycline alone. For their systemic overexpression of MAD2 a WT DOX and WT-DOX group were taken along and looking at the graphs in Figure 2 Dox itself clearly has an effect on absolute BM cell number, % of HSPCs and %CD19+B220+ cells. Authors do compare MR group to DOX control which is correct, but they should comment on effect of DOX alone. In line with this, in their experiments shown in Figure1 authors should include data on R control (treated with DOX).
- 3) For their competition reconstitution experiment authors need to show that both cell populations (R and MR) have the same viability in vivo without DOX induction to prove that effects are due to MAD2 overexpression. Both cell populations could differ in viability due to differences in the lines that cannot be explained by MAD2 overexpression. A no DOX control should be added to show that both cell lines make up app. 50% of the cells after the same time period.

Other comments:

Abstract:

- In the abstract it is mentioned that co-deletion of Bim/Bcl2l11, Bid, Puma/bbc3 and Noxa/Pmaip are tested, while results only show combination of BIM/NOXA, BIM or Noxa alone and co-deletion of Bid and Puma/Bbc3. To me it is unclear which combinations have been tested and what is shown in the Figure.
- Line 39 - 46: almost half of the abstract is introduction to the topic. Should be shortened so abstract gives readers more information about actual manuscript.

Introduction:

- Line 103: do authors mean breast cancer?
- Line 110: This sentence is not correct, should be checked
- Line 126: Although authors do check for aneuploidy, MAD2 overexpression did not influence aneuploidy in their model system. It should therefore be removed from this sentence as it is not the focus of the paper.

Results:

- Line 131: Authors should explain why they decide to focus on the haematopoietic cells, as it is now this choice seems random.
- Line 169 - 180: is there a possibility that GFP is toxic to the cells, thereby explaining the fact that they are outcompeted by cells that contain both GFP and MAD2 overexpression? Experiment should be carried out by co-culturing GFPonly cells and GFP+MAD2 cells.
- Lines 196-205: The implication that MR cells survive due to exogenous BCL2 overexpression are able to overcome SAC is also shown by increased M-exit Figure 1I, could be mentioned here as well
- Line 226: authors should elaborate on differences in effect on B cells = no difference and T cells = decrease.
- Line 232-235: Or the other way around, for some reasons more mature T & B cells are made, so therefore there is a loss of progenitors which need to be replenished?
- Line 289: was this significant?

Figures:

Figure 1:

- R control is missing, should be added at least for 1C and 1D.
- 1E and 1F: as mentioned previously co-culture should be done with control GFP-positive cells to exclude toxicity effects of GFP in culture.
- 1G: would be better to include label underneath, the grey color is not visible and the same for two of the groups.

Figure 4:

- Figure 4A: Time-line is not clear. When was Dox-food given and what was the real endpoint? So how long after start of experiment?

Figure 5:

- From the figure legend it is not clear what the treatments are exactly.

Figure S2:

- R control has high levels of cell death at all time-points as compared to no dox controls of other celltypes. How can you explain this? There seems to be a difference in cell death between cell lines already from the start (not due to MAD2/BCL2)

Figure S3:

- What does the # mean in B?
- S3F and O: seems to be clear effect of Dox on absolute SP cell number and % Ter19+CD71+

P-T-P Reply, Karbon et al. 2023-58131V1.**Referee #1:**

In this work by Karbon et al the in vivo impact of long term activation of the spindle assembly checkpoint is investigated. The authors achieve long term SAC activation by DOX induced overexpression of Mad2 in mice and observe bone marrow aplasia and intestinal atrophy. Furthermore the authors investigate how pro-apoptotic proteins contribute to in vivo phenotypes and find that BIM co-depletion prevents intestinal atrophy but not myelosuppression. Overall this is a very carefully conducted study that provides important insight into in vivo consequences of prolonged SAC activation. I have some minor suggestions that would in my opinion strengthen the impact of the work.

We want to thank this referee for his/her supportive comment

I think it is important that the authors state how much additional Mad2 is expressed. I think this information is in Fig. 1B but should be quantified. In Fig. 1B they should also probe for Cdc20 to establish how much additional MCC is likely generated.

We appreciate the desire to quantify the amount of additional MAD2 present in cells upon overexpression. We have tried to find exposure times where endogenous MAD2 can be detected with the signal for overexpressed MAD2 still in a linear range. But using conventional x-ray films we failed to find such a window (please see Figure EV11, in our revised version). So, in the end we can only conclude that MAD2 is strongly overexpressed, but fail to quantify it. We also aimed to re-probe the membranes in Fig. 1B with antibodies targeting CDC20 (Santa Cruz, E7, SC13162 and Abcam, AR12, ab190711), but suffered repeat set-backs, also after repeating the experiment. First, CDC20 has a molecular weight of 55kDA, interfering with the Ig heavy chain and the antibodies appear to recognize preferentially human, but not mouse CDC20, as tested by us using mouse embryonic fibroblasts. This contrasts what is stated on the respective data sheets, but the companies also only show western blots using human cell lines. In summary, these cells overexpress a massive amount of MAD2, a substantial fraction of which is in the closed active conformation and this suffices to promote mitotic arrest. How much CDC20 is sequestered, we cannot say, but it suffices to promote effective mitotic arrest (Figure 1I, movies S1-4).

The level of Mad2 over expression should be compared to some of the previous studies from Benezra that they compare their work to. Also how do these levels compare to the elevated levels of Mad2 reported in cancers? It could be that low levels of Mad2 expression is more detrimental than extremely high levels of over expression (which I think is what they obtain here).

This is a valid point, yet, we only maintain the *CoIA1-HA-MAD2* mice used in our study (PMID: 27292643) and not the original strain (PMID: 17189715), precluding a direct comparison side by side.

But, for this referee and review purposes only, we can show that mRNA and expression data are comparable between both mouse strains, at least in mammary organoids treated with doxycycline. These organoids were previously generated in the Sotillo lab, where both mouse strains were compared (Figure 1A, for review only). Hence, a direct comparison of protein levels can be found in the supplement of Rowald et al, 2016. (PMID: 27292643), shown here again, for this reviewers convenience (Figure 1B, for review only). This comparison indicates that protein levels are higher in the strain used by us, compared to that originally described, at least in mammary organoids. However, since we never analyze mammary gland tissue in our study, we believe it is not meaningful to integrate this data set in our manuscript. We discuss this observation by Rowald et al (PMID: 27292643) in more detail on page 17.

To address if the levels of *Mad2* mRNA in our transgenic mice are similar to those found in human cancer, we have also extracted a comparison between human breast cell lines and KRAS/MAD2 breast cancers from animal cohorts published in 2016 (PMID: 27292643). This data suggests that indeed the levels of *MAD2* in human breast cancer cell lines and primary tumors in these mice are comparable (Figure 1C). As we do not see spontaneous tumor development and have not analyzed mammary tissue in our study we can only refer to potential tissue type dependent differences in the discussion, again on page 17.

Figure for referee with unpublished data and its description has been removed upon request by the authors.

Is it possible for the authors to investigate cellular fate in relation to GFP expression (proxy for Mad2 levels) to address this?

Indeed we correlate expression of GFP with that of MAD2, without properly demonstrating that expression levels are indeed proportional. In an effort to address this, we have isolated three populations of HOXB8 cells based on GFP expression, i.e. GFP-high, -low and -negative, by cell sorting, to correlate GFP with MAD2 transgene expression in western blot analysis (Figure EV1H).

This revealed that while we obtained a clear correlation for GFP, the MAD2 transgene was expressed at similar levels in GFP-high vs -low cells alike and, upon longer exposure also detectable in GFP-negative cells. This indicates that we actually underestimate the levels of transgenic MAD2 in our analyses presented in Figure 1 and EV1, when we use GFP as a proxy for MAD2. Of note, even prolonged exposure times did not allow us to detect a GFP signal in the GFP-negative population shown in Figure EV1H. We refer to this in the results on page 7. The problem with correlating GFP expression

levels to cell fate, i.e. cell death, is that cells that undergo apoptosis also lose GFP expression very quickly. This may relate to the observation that GFP-negative cells still expressed low levels MAD2. GFP alone, however, did not prove toxic to these cells, as questioned by referee # 3 (Figure EV1H).

Small typos:

be consistent in either cyclin or Cyclin and securin or Securin

page 3 line 67: securing should be securin

page 16 line 419: above - do they mean "consistent with the later"?

Thank you for spotting these inconsistencies that are now corrected.

Referee #2:

Karbon et al have examined spindle assembly checkpoints in an interesting way by assessing if mitochondrial apoptosis limits the survival of cells with chronic SAC activity in order to prevent aneuploidy and CIN-related pathology. The authors further identify the pro-apoptotic protein BIM as rate limiting for apoptosis induction in the GI tract in response to SAC perturbation.

We are pleased to hear this referee finds our work "interesting"

1. The authors state lines 186-187, "that the presence of BCL2 reduced cell death in mitosis (diM) from about 60% to 20% in MR2 cells", was this statistically significant and if so, this should be stated and shown in Figure 1L.

We have conducted statistical analysis using an unpaired t-test, confirming significance and added the information to the figure legend, adhering to journal style.

2. Relating to figure 2, no karyotypic abnormalities were detected in the presence or absence of Dox in R, R2, MR or MR2 cells nor within live MR2 cells remaining at days 3 or 7, perhaps the optimal timepoint was not assessed and 12 h or 2 days may have been optimal, since all cells were dead at day 3.

HoxB8 progenitor cells duplicate within less than a day and hence we were considering 24h as a suitable timepoint to catch aneuploidies, working with asynchronous cultures, as MAD2 transgene expression was readily detectable already after 12h (Figure 1A). Importantly, at 24h cell death was not yet initiated on a broad scale, when compared to untreated, while at 48h cell death was already very prominent in MR cells on Dox, expressing MAD2 (Figure 1D). As such, we are confident that we do not miss the window where "transient" aneuploidies may arise. Admittedly, we were also surprised by these results, but reasoned that if cells were to mis-segregate chromosomes frequently, we should catch this at 24h, or at least on the BCL2 transgenic background, where cell death is negligible at this time point (Figure 1D). There is also no reason why such aneuploid cells should be diluted out over time, as they actually stop proliferating if unable to die. So, we are confident we did not miss anything,

3. The bone marrow reconstitution experiments (Figure 3, 4) lack positive controls, such as reconstitution with the respective parental BM cells for comparative studies, i.e. mice which have also undergone irradiation and reconstitution. Are any of the values in Figure 3 significantly altered compared to T0 (50:50)?

We indicate significant differences and p values to Figure 3 and legend. We are not sure, however, if we fully understand this comment fully, i.e. which type of positive controls this referee deems missing. We have mixed bone marrow isolated from rtTA (R) and rtTA/MAD2 transgenic mice (MR) mice. If this referee was referring to a control using wild type vs. rtTA-derived bone marrow placed on Doxycycline,

we argue that, yes, it is possible that Doxycycline treatment affects the reconstitution potential of bone marrow stem cells, but such a “systematic bias” would also be present in our experimental groups, where all recipient mice receive Dox-food, and both groups of donor cells express the reverse transactivator (rtTA). Hence, we believe it is fair to say that effects observed are due to MAD2 transgene expression.

The same holds true for Figure 4, where all bone marrow donor cells express the *rtTA* allele, as well as the GFP reporter, allowing us to selectively characterize consequences of MAD2 overexpression in the absence and presence of the BCL2 transgene, which is also controlled for individually in RG2 recipient mice.

4. Figure 3A the authors indicate in lines 243-244 that “pre-B and myeloid colony formation was drastically impaired upon Dox treatment”, what does drastically mean, was the difference significant as no significance values are indicated in Figure 3A.

As almost no myeloid colonies form when MAD2 is overexpressed by Doxycycline in the presence of MCSF or GCSF (Figure 3A), we believe that “drastic” is not an exaggeration ($p < .001$), however, we now phrase it differently as “strongly reduced”. We apologize the oversight of omitting p values that are now provided in figure and the corresponding information in the legend.

5. Figure 4A appears to be lacking all annotation.

We have completed this figure legend and provided the missing information.

6. Whilst dissecting the consequences of MAD2 overexpression on the haematopoietic system, the authors noted a reduction thymic cellularity. The dissection of the proportional difference in the earlier T cell progenitors was fairly rudimentary and requires further investigation, particularly since some the subpopulations are cycling while others non-cycling, i.e. staining with CD25 and CD44 to dissect out the DN 1-4 stages in T cell maturation most affected by MAD2 overexpression and thus the proliferative capacity of the progressive stages of T cell development.

This is a fair point, luckily we had already acquired this data, but omitted it in the first submission for simplification. We are happy to include this data now, indicating a loss of DN3 thymocytes and an relative increase in DN1/4 stages, in the revised version of Figure 2M, N. Results are presented on page 9.

7. Figure S6A-E, why is the 56 day time point data not integrated into the linear curve of each these graphs. Clearly something has happened at this time point, with a drop in all parameters analysed, yet no explanation is offered. Figure S6J, were any of the graphed values significant, if not, indicate in the text, rather than “comparable”?

This referee is correct in pointing out that the percentage of GFP+ cells on day 56, originating from our first cohort of reconstituted mice, shown in Figure 4 and EV3 (former S5), and displayed again for reasons of comparison in EV3, is on average lower than in our second cohort, set up for long-term cohort analysis. The reason for this remains obscure. Regardless, we did actually also analyze this second cohort on day 56 and display this now in the revised Figure EV3. We initially did not include these data, as it did not change the slope of the curve and we thought it facilitated data visualization. We included the percentage of GFP+ cells in the peripheral blood from the first cohort, analyzed on day 56, again in EV3 only as a reference to document biological variation between the two experimental cohorts (indicated in the legend).

Unexpectedly, the levels of reporter expression still increased over time, which we did not expect, based on our initial analysis shown in Figure 4B. In any case, we were not aiming for a direct comparison between cohorts, but wanted to monitor the long term consequences of MAD2 overexpression in hematopoietic cells. Please note that it is also only the percentage of GFP+ cells that differs between cohorts on day 56, not the percentage of leukocytes in peripheral blood. Importantly, our data indicates

that transgene induction was actually more effective in our second cohort that we monitored for tumor development for up to 12 month.

8. Line 333-334, the authors indicate in relation to the MAD2 overexpression none of the animals' developed signs of malignant disease" and lines 340-341 "still MAD2 overexpression appears insufficient to promote spontaneous malignancies within 12 months" what signs were assessed, which organs, which pathology performed, there are no data to support these statements.

We believe we have not phrased our conclusion well. In fact, we have monitored the composition of peripheral blood, primary and secondary lymphatic organs, using a broad panel of leukocyte markers and flow cytometry, now provided as supplementary Table 1 in the Appendix. Based on this analysis, the composition of the hematopoietic compartment and leukocyte subset distribution in reconstituted animals was found comparable to that of non-reconstituted animals at the age of two month (previously shown in Figure S6, now Appendix Figure 3). No expansion of a distinct population of myeloid cells or lymphocytes was observed by flow cytometry, nor did we observe abnormal white blood cell counts, splenomegaly, enlarged thymus or lymph nodes. A histopathological analysis based on H&E staining of tissue sections was not conducted. However, animals did not show any signs of burden, based on routine weekly visual inspection by animal caretakers and the first author of this study before sacrifice. We did not assess non-hematopoietic tissues, as transgene expression was restricted to blood cells in the reconstitution setting. We now better explain this in the result section on page 14 and hope this clarification will be satisfactory.

Referee #3:

Karbon et al. aimed to describe consequences of apoptosis after MAD2 overexpression. They describe that overexpression of MAD2 leads to a mitotic delay and cell-death, which can be reduced by expression of the pro-survival protein BCL2. Although systemic overexpression of MAD2 is not viable, authors were able to study effects of MAD2 overexpression in vivo by studying effects on hematopoietic system, by making use of a very elegant competitive reconstitution experiment. Mad2 overexpression led to myeloablation which was rescued by BCL2 overexpression, identifying a role for apoptosis in this process. Next to that, they have shown that SAC perturbation causes severe colitis and gastrointestinal syndrome which may have been a cause for the rapid death of mice with systemic overexpression of MAD2. Authors identified the pro-apoptotic BIM as an essential factor leading to decreased survival of the transgenic mice. However, deleting BIM did not lead to a rescue of the MAD2 overexpression phenotype in the hematopoietic system. At this point it is not clear why these effects differ in each organ system or which other factors play a role in this.

The manuscript describes interesting results which could contribute to understanding the effect of SAC perturbation on the hematopoietic system and GIT, however, some major concerns regarding the manuscript are listed below.

We want to thank this referee for his/her positive comments and for considering our results as "interesting"

Main comments:

1) The authors set out to study the molecular players mediating cell death in response to chronic SAC deregulation in vivo. Although authors do study this partly, their manuscript would greatly benefit from a more elaborate study of additional apoptotic effectors to truly understand the role of apoptosis after MAD2 overexpression.

Where BIM is an essential factor regulating effects on GIT, it does not have an effect on the hematopoietic system, where overexpression of BCL2 had a rescue effect. Why did the authors not

study effect of BCL2 in the GIT? (In addition to that it is very unclear which factors and combinations were effectively tested, see comment below).

Unfortunately, we do not have a transgenic mouse model at hand that would allow overexpression of transgenic BCL2, or other pro-survival proteins for that matter, in the gastrointestinal tract. I am also not aware that such model is currently available. Our BCL2 transgenic mice do however allow transgene expression in all blood cell lineages and confers an even more profound cell death resistance, as compared to transgenic MCL1 (PMID: 20631380, PMID: 29498802). Nonetheless, we believe that results overexpressing either anti-apoptotic protein as a transgene will yield similar results and would not tell us, which of the two proteins was relevant endogenously, as only deletion of either gene could achieve that. Loss of MCL1 is early embryonic lethal and deletion in the hematopoietic system abrogates blood cell formation (PMID: 15718471), precluding analysis.

As an alternative approach, analyzing animals lacking different BH3 only proteins was the best option available to us, to assess the role of mitochondrial apoptosis in the GI tract in response to MAD2 overexpression. I would like to emphasize that while overexpression of anti-apoptotic proteins can be seen as rather artificial due to high exogenous protein levels, studying consequences of the loss of BH3 only proteins does not suffer such limitation. Hence, we comprehensively explored the role of 4 out of 8 BH3-only proteins involved in cell death (PMID: 27083995).

In their discussion authors mention that differential effect of BIM and NOXA on hematopoietic system may be due to other apoptotic effectors BMF and BID, (also MCL1?), which should be tested in order to clarify this discrepancy and to answer their research question.

We appreciate this concern but if we read the comment correctly, this reviewer suggests to test the impact of loss of BID or BMF on cell death induced by MAD2 overexpression. Loss of BID has actually been analyzed and the data was included in the first submission. We show in Figure 6 and EV5 (former S8) that combined loss of BID and PUMA does not rescue hematopoietic nor GI-tract phenotypes in vivo. This is now further confirmed by analyzing HOXB8 cells ex vivo lacking both BH3 only proteins. Double-mutant cells die at rates comparable to wild type cells (EV5, former S8). So, we believe it is fair to conclude that BID does not play a role, neither in the hematopoietic system nor GI tract.

Indeed, we cannot not exclude a role for BMF in this process, but we no longer maintain *Bmf*^{-/-} mice in our facility. BMF is mainly expressed in B cells, but no other hematopoietic cell types in steady state (PMID: 24632712). Accordingly, we failed to detect BMF by western blot analysis in MAD2 transgenic HOXB8 cells, neither at base-line nor in upon doxycycline treatment that promotes cell death, excluding a role in HOXB8 cells (Appendix, Figure 4B).

However, we noted that the longer isoform of BMF, transcribed in mice from a non-conventional start site (PMID 0706276) appears to be induced / stabilized in the GI tract in response to MAD2 overexpression. This result is now shown in the Appendix, Figure 4A. It is unclear though if this signal originates from epithelial cells, or infiltrating leukocytes. Hence, BMF may contribute to the tissue damage seen in the GI tract, discussed on page 14. However, we have previously shown that loss of BMF does synergize with loss of BIM in the B cells, providing higher degree of cell death protection, e.g. to glucocorticoids, or ligation of the B cell receptor, mimicking interaction with self-antigens (PMID: 24632712). As loss of BIM already suffices to rescue these animals from the lethal effects of MAD2 expression, a contribution of BMF is hard to delineate. We can also not be sure that the expression we see originates in epithelial cells, and may be from infiltrating hematopoietic cells. As such, we believe that BMF is not relevant in the GI-tract, but we acknowledge published literature implicating it in cell death upon SAC activation (already cited in version 1) and that it may add to the loss of hematopoietic cells, that cannot be restored by any of the BH3-only proteins tested (discussion, page 20).

2) For their experiments authors make use of doxycycline which is a well-known anti-inflammatory agent. Authors fail to accurately describe effects of doxycycline alone.

For their systemic overexpression of MAD2 a WT DOX and WT-DOX group were taken along and looking at the graphs in Figure 2 Dox itself clearly has an effect on absolute BM cell number, % of HSPCs and %CD19+B220+ cells.

Authors do compare MR group to DOX control which is correct, but they should comment on effect of DOX alone. In line with this, in their experiments shown in Figure1 authors should include data on R control (treated with DOX).

We agree that these controls are important. Regarding Figure 2, even though it seems that DOX treatment alone affects BM cell number, statistical analysis using 1-way ANOVA failed to reveal significant differences across conditions and genotypes. Even a direct comparison of WT ± DOX using an unpaired t-test does not show a significant difference.

Regarding Figure 1, the control this reviewer asks for can be found in EV5B, grey bars (former S8) where R cells have been treated also with Dox (grey bars), but this did not impact on cell survival, and the increase in cell death seen after 72h is statistically not significant (1-way ANOVA) and the variation seen most certainly due to overgrowth of these cells within 72h. We will now include a cross-reference in the text and state that Dox treatment alone has no effect on viability.

3) For their competition reconstitution experiment authors need to show that both cell populations (R and MR) have the same viability in vivo without DOX induction to prove that effects are due to MAD2 overexpression.

Both cell populations could differ in viability due to differences in the lines that cannot be explained by MAD2 overexpression. A no DOX control should be added to show that both cell lines make up app. 50% of the cells after the same time period.

We do honor the concern of this referee, but we believe that this is very likely not a confounding issue.

First, we have transplanted primary bone marrow from rtTA and rtTA_MAD2 mice in a 1:1 ratio into recipient mice that lack own blood cells due to radiation-mediated preconditioning; meaning, we did not use "cell lines" that may differ due to clonal variation selected in tissue culture.

Second, even if we conduct the proposed experiment and would note a drop in cells originating from rtTA_MAD2 transgenic bone marrow in the absence of DOX, this effect would likely be due to leaky transgene expression, rather than an undefined "off target effect" due to a hemizygous SNP or SNV introduced by crossing in the MAD2 allele on the rtTA background. The MAD2 expression cassette has been targeted into the *Col-1a* locus, which is seen as a safe-harbor location and does not bare the risk of traditional approaches where transgenes were inserted randomly into the mouse genome affecting nearby genes.

Finally, we are truly confident that the risk pointed out by this referee is minimal, as we also show in that hematopoietic stem cells isolated from rtTA and rtTA_MAD2 mice display comparable clonogenic potential in the absence of DOX treatment (Figure 3A).

Other comments:

Abstract:

- In the abstract it is mentioned that co-deletion of Bim/Bcl2l11, Bid, Puma/bbc3 and Noxa/Pmaip are tested, while results only show combination of BIM/NOXA, BIM or Noxa alone and co-deletion of Bid and Puma/Bbc3. To me it is unclear which combinations have been tested and what is shown in the Figure.

Will have rephrased the abstract to be more clear.

- Line 39 - 46: almost half of the abstract is introduction to the topic. Should be shortened so abstract gives readers more information about actual manuscript.

We have made an attempt to reduce background information in the abstract to better highlight our findings also honoring the abstract limit of 175 words now.

Introduction:

- Line 103: do authors mean breast cancer?

Yes, thanks for spotting this error

- Line 110: This sentence is not correct, should be checked

Corrected

- Line 126: Although authors do check for aneuploidy, MAD2 overexpression did not influence aneuploidy in their model system. It should therefore be removed from this sentence as it is not the focus of the paper.

We have adapted this sentence to:apoptosis limits the survival of cells experiencing chronic SAC activity and potentially CIN-related pathology.

Results:

- Line 131: Authors should explain why they decide to focus on the haematopoietic cells, as it is now this choice seems random.

We now rephrased this sentence to provide rational. *First, we assessed the impact of aberrant MAD2 expression on the survival of highly proliferative cells within the haematopoietic system, expecting them to be highly vulnerable to SAC perturbation.*

- Line 169 - 180: is there a possibility that GFP is toxic to the cells, thereby explaining the fact that they are outcompeted by cells that contain both GFP and MAD2 overexpression? Experiment should be carried out by co-culturing GFPonly cells and GFP+MAD2 cells.

This point is certainly valid. However, the experiment proposed by this referee cannot be conducted, as we would no longer be able to discriminate both genotypes in the same dish, if both genotypes are positive for GFP (co-culturing GFPonly and GFP+MAD2 cells). However, we have tested if Dox-induced expression of GFP in HoxB8 RG cells, expressing the rtTA to drive GFP, increases cell death over that seen in R control cells only expressing the rtTA. In flow cytometric analyses, we did not find a negative effect of GFP expression on cell survival, now shown in EV11.

- Lines 196-205: The implication that MR cells survive due to exogenous BCL2 overexpression are able to overcome SAC is also shown by increased M-exit Figure 1I, could be mentioned here as well

Has been highlighted again

- Line 226: authors should elaborate on differences in effect on B cells = no difference and T cells = decrease.

We believe this must be a misunderstanding – indeed, we see a clear drop in thymocyte number, a depletion of CD4+8+ thymocytes and a relative increase in the percentage of single positive cells in this tissue. Mature T cells in the periphery were not affected, because they are long lived and not actively

cycling. Indeed, mature recirculating B cells increase in the bone marrow while progenitors are depleted, despite overall BM cellularity not being significantly altered. This suggests that homing/retention of recirculating B cells may be enhanced in response to depletion of progenitor B cells, e.g. due to freed space in bone marrow niches, which we mention now in the text on page 9.

- Line 232-235: Or the other way around, for some reasons more mature T & B cells are made, so therefore there is a loss of progenitors which need to be replenished?

We are not entirely sure if we understand this comment. Data in Fig S3 of the initial submission (now appendix Figure 1) shows that the percentage of mature IgM+D+ B cells in spleen only slightly increases, while mature T cells do not significantly change (Appendix 1H). No effect is seen in lymph nodes for both T and B cells (Appendix 1K). The increase in mature B cells may add to the increase of recirculating B cells in bone marrow, but the reason for this phenomenon remains unclear.

- Line 289: was this significant?

When performing an ANOVA analysis, MRG and MRG2 cells do not differ significantly from each other, but when conducting a t-test comparing data on day 10 and day 56 separately, yes. My personal understanding is that this selective comparison is actually not correct, as we have different variables to consider, genotype, GFP-expression, as well as time.

- R control is missing, should be added at least for 1C and 1D.

We apologize for the confusion. This control can be found in EV5B, lower panel, grey bars (former Figure S8), as we wanted to minimize complexity in the primary figure panels. There is no negative impact of DOX treatment on cell survival in the first 48h and only a mild increase at 72h is seen, likely due to overcrowding in culture. We will cross-reference this data set to point this out.

- 1E and 1F: as mentioned previously co-culture should be done with control GFP-positive cells to exclude toxicity effects of GFP in culture.

We can exclude GFP toxicity, as mentioned above. Corresponding data shown in EV11.

- 1G: would be better to include label underneath, the grey color is not visible and the same for two of the groups.

This has been changed.

Figure 4:

- Figure 4A: Time-line is not clear. When was Dox-food given and what was the real endpoint? So how long after start of experiment?

We have optimized the cartoon and legend; we hope it is clear now.

Figure 5:

- From the figure legend it is not clear what the treatments are exactly.

We apologize for this shortcoming. The figure legend has been updated to better describe figure content.

Figure S2:

- R control has high levels of cell death at all time-points as compared to no dox controls of other cell types.

How can you explain this? There seems to be a difference in cell death between cell lines already from the start (not due to MAD2/BCL2)

Indeed, we often see such variation in HoxB8 cultures, but the relative increases are the important parameter to look for.

Figure S3:

- What does the # mean in B?

The hashtag symbol has been used as an abbreviation for “number” of cells. The labelling has been changed to be more clear.

- S3F and O: seems to be clear effect of Dox on absolute SP cell number and % Ter19+CD71+

The differences in S3F/O (now Appendix Fig. 1) are statistically significant, by ANOVA or t-test analyses.

Dear Prof. Villunger,

Thank you for the submission of your revised manuscript to our editorial offices. I have now received the reports from the three referees that I asked to re-evaluate the study, you will find below. As you will see, they now fully support the publication of the study in EMBO reports.

Before formal acceptance, I have these editorial requests I ask you to address in a final revised manuscript:

- Please provide a final title with not more than 100 characters (including spaces).
- Please provide the abstract written in present tense throughout.
- Please remove the ORCID IDs from the title page. Please link the ORCID IDs to the author profiles in our submission system (if not already done). Please find instructions on how to link the ORCID ID to the account in our manuscript tracking system in our Author guidelines: <http://www.embopress.org/page/journal/14693178/authorguide#authorshipguidelines>
- Please remove the list of abbreviations from the title page and make sure each abbreviation is defined the first time mentioned in the manuscript text.
- We updated our journal's competing interests policy in January 2022 and request authors to consider both actual and perceived competing interests. Please review the policy <https://www.embopress.org/competing-interests> and update your competing interests if necessary. Please name this section 'Disclosure and Competing Interests Statement' and put it after the Acknowledgements section.
- We now use CRediT to specify the contributions of each author in the journal submission system. CRediT replaces the author contribution section. Please use the free text box to provide more detailed descriptions and do NOT provide your final manuscript text file with an author contributions section. See also our guide to authors: <https://www.embopress.org/page/journal/14693178/authorguide#authorshipguidelines>
- Please acknowledge BioRender in the acknowledgement section.
- Please reduce the number of keywords to 5 and order the manuscript sections like this using these names: Title page - Abstract - Keywords - Introduction - Results - Discussion - Methods - Data availability section - Acknowledgements - Disclosure and Competing Interests Statement - References - Figure legends - Expanded View Figure legends
- Please make sure that the number "n" for how many independent experiments were performed, their nature (biological versus technical replicates), the bars and error bars (e.g. SEM, SD) and the test used to calculate p-values is indicated in the respective figure legends (for main, EV and Appendix figures) of the final revised manuscript. Please also check that all the p-values are explained in the legend, and that these fit to those shown in the figure. Please provide statistical testing where applicable. Please avoid the phrase 'independent experiment', but clearly state if these were biological or technical replicates. Please also indicate (e.g. with n.s.) if testing was performed, but the differences are not significant. In case n=2, please show the data as separate datapoints without error bars and statistics. See also:

<http://www.embopress.org/page/journal/14693178/authorguide#statisticalanalysis>

If $n < 5$, please show single datapoints for diagrams. It seems that presently some diagrams show only partial statistics or miss the 'n.s.'. Moreover:

- Please note that in figures 1c-g, i; EV 3k; there is a mismatch between the annotated p values in the figure legend and the annotated p values in the figure file that should be corrected.
- Please note that the box plots need to be defined in terms of minima, maxima, centre, bounds of box and whiskers, and percentile in the legends of figures 1g; 2a; 5a, d-f, h; 6a; EV 3f; EV 4e.
- Please note that information related to n is missing in the legends of figures 5f, h; EV 1f; EV 3a-f, j-k.
- Although 'n' is provided, please describe the nature of entity for 'n' in the legends of figures 2n; 3a, e; 5d-e; EV 5a.
- Please note that the error bars are not defined in the legends of figures 4f, g-j; EV 1f.
- Please add to each legend (main, EV and Appendix figures) a 'Data Information' section explaining the statistics used or providing information regarding replicates and scales. See:

- Please add scale bars of similar style and thickness to microscopic images, using clearly visible black or white bars (depending on the background). Please place these in the lower right corner of the images themselves. Please do not write on or near the

bars in the image but define the size in the respective figure legend. Presently, most scale bars are hard to see and have text nearby. Please check.

- Per journal policy, we do not allow 'data not shown', which is stated in the manuscript (page 9). All data referred to in the paper should be displayed in the main or Expanded View figures, or an Appendix. Thus, please add these data (or change the text accordingly if these data are not central to the study). See:
<https://www.embopress.org/page/journal/14693178/authorguide#unpublisheddata>

- Please make sure that all the funding information is also entered into the online submission system and is complete and similar to the one in the manuscript text file (in the Acknowledgements). Presently, the grants DFG TRR353 initiative and DOC fellowship program of the Austrian Academy of Sciences are only mentioned in the Acknowledgements.

- Please name the EV figures 'Figure EVx' (not just EVx) and adjust the callouts accordingly.

- The nomenclature of the Appendix figures and their callouts needs correction. The names should be "Appendix Figure Sx". Please check.

- Please upload the information provided in the Appendix tables as reagents and tools table (and remove the tables from the final Appendix). I have attached templates for that in word or excel format. Please upload the filled in table to the manuscript tracking system as 'Reagent Table' file. Please also adjust any callouts to this information. The example linked below shows how the table will display in the published article and includes examples of the type of information that should be provided for the different categories of reagents and tools. Please list your reagents/tools using the categories provided in the template and do not add additional subheadings to the table. Reagents/tools that do not fit in any of the specific categories can be listed under "Other":

https://www.embopress.org/pb%2Dassets/embo-site/msb_177951_sample_FINAL.pdf

- Please upload the movie zipped together with their legends (as readme.txt file), 1 folder per movie. Please remove the movie legends from the manuscript text file.

- There is a table on page 27. Please name this Table 1 and add a title and call-outs in the manuscript text.

- Thanks for providing a link to the source data. Please upload the filled in SD checklist (attached) with your final submission and make sure that all the source data is provided (uploaded to BioStudies) as requested.

Best,

Referee #1:

The authors have done their best to address my concerns and I think the paper should be published. Overall this is a nice and solid story.

Referee #2:

Karbon et al - revision

1. The authors state lines 186-187, "that the presence of BCL2 reduced cell death in mitosis (diM) from about 60% to 20% in MR2 cells", was this statistically significant and if so, this should be stated and shown in Figure 1L.

The authors have provided the additional data requested.

2. Relating to figure 2, no karyotypic abnormalities were detected in the presence or absence of Dox in R, R2, MR or MR2 cells nor within live MR2 cells remaining at days 3 or 7, perhaps the optimal timepoint was not assessed and 12 h or 2 days may have

been optimal, since all cells were dead at day 3.

The reviewer is satisfied with the explanation provided.

3. The bone marrow reconstitution experiments (Figure 3, 4) lack positive controls, such as reconstitution with the respective parental BM cells for comparative studies, i.e. mice which have also undergone irradiation and reconstitution. Are any of the values in Figure 3 significantly altered compared to T0 (50:50)?

The reviewer is satisfied with the explanation provided.

4. Figure 3A the authors indicate in lines 243-244 that "pre-B and myeloid colony formation was drastically impaired upon Dox treatment", what does drastically mean, was the difference significant as no significance values are indicated in Figure 3A.

The authors have provided the additional data requested.

5. Figure 4A appears to be lacking all annotation.

Figure looks better now.

6. Whilst dissecting the consequences of MAD2 overexpression on the haematopoietic system, the authors noted a reduction thymic cellularity. The dissection of the proportional difference in the earlier T cell progenitors was fairly rudimentary and requires further investigation, particularly since some the subpopulations are cycling while others non-cycling, i.e. staining with CD25 and CD44 to dissect out the DN 1-4 stages in T cell maturation most affected by MAD2 overexpression and thus the proliferative capacity of the progressive stages of T cell development.

The authors have provided the additional data requested and it looks impressive.

7. Figure S6A-E, why is the 56 day time point data not integrated into the linear curve of each these graphs. Clearly something has happened at this time point, with a drop in all parameters analysed, yet no explanation is offered. Figure S6J, were any of the graphed values significant, if not, indicate in the text, rather than "comparable"?

8.

The reviewer is satisfied with the explanation provided.

9. Line 333-334, the authors indicate in relation to the MAD2 overexpression none of the animals' developed signs of malignant disease" and lines 340-341 "still MAD2 overexpression appears insufficient to promote spontaneous malignancies within 12 months" what signs were assessed, which organs, which pathology performed, there are no data to support these statements.

10.

11. The reviewer is satisfied with the explanation provided.

For the Editor

Overall, the authors have made addressed the reviewers concerns by supplying the requested data or providing additional clarity.

Referee #3:

I am satisfied with the response of the authors to my comments, especially regarding the controls, as well as the changes they have made to clarify some things. I still think the manuscript could benefit from a more elaborate study in the different apoptotic factors to understand how these processes exactly work in the different organ systems, but I also understand that this maybe goes beyond the scope of this paper. Therefore, in my opinion the manuscript is now suitable for publication.

All editorial and formatting issues were resolved by the authors.

Prof. Andreas Villunger
Medical University of Innsbruck
Institute of Dev. Immunology
Biocenter
Innrain 80
Innsbruck A-6020
Austria

Dear Prof. Villunger,

I am very pleased to accept your manuscript for publication in the next available issue of EMBO reports. Thank you for your contribution to our journal.

Yours sincerely,
